# Lysosome activity is modulated by multiple longevity pathways and is important for lifespan extension in *C. elegans*

Yanan Sun[1,2,3], Meijiao Li[4], Dongfeng Zhao[3], Xin Li[3], Chonglin Yang[4], Xiaochen Wang[3,5]*

[1]College of Life science, Beijing Normal University, Beijing, China; [2]National Institute of Biological Sciences, Beijing, China; [3]National Laboratory of Biomacromolecules, CAS Center for Excellence in Biomacromolecules, Institute of Biophysics, Chinese Academy of Sciences, Beijing, China; [4]State Key Laboratory of Conservation and Utilization of Bio-Resources in Yunnan, and Center for Life Sciences, School of Life Sciences, Yunnan University, Kunming, China; [5]College of Life Sciences, University of Chinese Academy of Sciences, Beijing, China

*For correspondence:
wangxiaochen@ibp.ac.cn

**Abstract** Lysosomes play important roles in cellular degradation to maintain cell homeostasis. In order to understand whether and how lysosomes alter with age and contribute to lifespan regulation, we characterized multiple properties of lysosomes during the aging process in *C. elegans*. We uncovered age-dependent alterations in lysosomal morphology, motility, acidity and degradation activity, all of which indicate a decline in lysosome function with age. The age-associated lysosomal changes are suppressed in the long-lived mutants *daf-2*, *eat-2* and *isp-1*, which extend lifespan by inhibiting insulin/IGF-1 signaling, reducing food intake and impairing mitochondrial function, respectively. We found that 43 lysosome genes exhibit reduced expression with age, including genes encoding subunits of the proton pump V-ATPase and cathepsin proteases. The expression of lysosome genes is upregulated in the long-lived mutants, and this upregulation requires the functions of DAF-16/FOXO and SKN-1/NRF2 transcription factors. Impairing lysosome function affects clearance of aggregate-prone proteins and disrupts lifespan extension in *daf-2*, *eat-2* and *isp-1* worms. Our data indicate that lysosome function is modulated by multiple longevity pathways and is important for lifespan extension.

## Introduction

Lysosomes are dynamic organelles responsible for macromolecule degradation and catabolite recycling. Lysosomes also serve as a signaling hub to integrate nutritional, energy and growth factor information and coordinate cellular responses through key regulatory modules docked on the lysosomal surface (*Lawrence and Zoncu, 2019*). By acting as centers of degradation, recycling and signaling, lysosomes play crucial roles in a variety of fundamental processes to maintain cell and tissue homeostasis. Lysosomal dysfunction is associated with a number of age-related pathologies, which suggests the importance of lysosome function in the aging process (*Carmona-Gutierrez et al., 2016*).

Aging is considered as a process of gradual deterioration of physiological functions that leads to decreased survival and increased risk of death (*López-Otín et al., 2013*). One of the most universal hallmarks of aging is the decline in protein homeostasis (*López-Otín et al., 2013*). Studies in a variety of organisms have uncovered age-dependent accumulation of misfolded and damaged proteins,

which may impair cell function and homeostasis, leading to the development of age-related diseases (*Cuervo and Dice, 2000*; *Terman and Brunk, 2004*; *Martinez-Vicente et al., 2005*). Misfolded, aggregated and damaged proteins can be removed by the proteasome or cleared through the autophagy-lysosome pathway. As the key organelle for cellular degradation, lysosomes exhibit age-related changes such as increased size, number and content; increases and decreases in lysosomal hydrolase activity have also been reported (*Truschel et al., 2018*; *Sarkis et al., 1988*; *Cuervo and Dice, 1998*; *Hayflick, 1980*; *Bolanowski et al., 1983*; *Yoon et al., 2010*; *Cuervo, 2010*). Moreover, vacuolar acidity reduces during replicative aging in budding yeast, and lysosomal pH appears to increase with age in the *C. elegans* intestine (*Hughes and Gottschling, 2012*; *Baxi et al., 2017*). In addition, there is evidence for increased lysosomal gene expression with age, which is considered as a compensatory response to altered protein homeostasis (*de Magalhães et al., 2009*; *Cellerino and Ori, 2017*). Therefore, the causal connection between age-associated lysosomal changes and accumulation of abnormal proteins remains unclear.

Like many other biological processes, the aging process is subjected to regulation. Intrinsic and extrinsic longevity regulatory pathways have been identified that play evolutionarily conserved roles. One such pathway is the insulin/IGF-1 signaling (IIS) pathway, which controls aging in *C. elegans*, insects and mammals, and extends the lifespan of these organisms when attenuated (*Anisimov and Bartke, 2013*). In worms, reducing IIS, such as through mutation in the *daf-2* gene, which encodes the sole *C. elegans* insulin/IGF-1 receptor, leads to significantly increased adult longevity (*Kenyon et al., 1993*). The extension of longevity by reduced IIS involves a phosphorylation cascade that ultimately results in nuclear translocation of the DAF-16/Forkhead box (FOXO) and the SKN-1/Nuclear factor-erythroid-related factor 2 (NRF2) transcription factors and subsequent transcriptional regulation of their target genes (*Murphy and Hu, 2013*; *Tullet et al., 2008*). DAF-16 and SKN-1 have both distinct and overlapping functions in lifespan extension under the condition of reduced IIS (*Tullet et al., 2008*; *Ewald et al., 2015*). The heat-shock transcription factor HSF-1 also acts downstream of the IIS pathway. HSF-1 may collaborate with DAF-16 to regulate the expression of chaperone genes, thus contributing to the longevity of *daf-2* mutants (*Hsu et al., 2003*). In addition to down-regulation of the IIS pathway, increased longevity can be achieved by reducing food intake or impairing mitochondrial function. Both caloric restriction and mild inhibition of mitochondrial respiration extend the lifespan of many organisms (*Kenyon, 2010*). In worms, the feeding-defective *eat-2* mutation significantly lengthens the lifespan, and this requires the function of PHA-4/FOXA and SKN-1/NRF2 transcription factors (*Lakowski and Hekimi, 1998*; *Panowski et al., 2007*; *Park et al., 2010*). Reducing mitochondrial function may produce a low dose of stressors such as reactive oxygen species (ROS), which elicit protective adaptive responses and induce pro-longevity effects through DAF-16, SKN-1 and the hypoxia-inducible factor HIF-1 (*Ventura, 2017*; *Senchuk et al., 2018*; *Schmeisser et al., 2013*; *Lee et al., 2010*; *Yang and Hekimi, 2010*). The different longevity regulatory pathways are not completely independent but may utilize overlapping mechanisms as they share downstream transcription factors.

Consistent with the evidence that declining protein homeostasis serves as an aging marker, long-lived worms can preserve their proteome with age. Several hundred proteins with diverse functions have been identified that become more insoluble with age in wild-type *C. elegans* (*David et al., 2010*). The increased protein insolubility and aggregation, however, is significantly delayed or even halted in long-lived *daf-2* worms (*David et al., 2010*). The mechanisms by which *daf-2* mutants maintain protein homeostasis are not fully understood. In *daf-2* animals, there is increased autophagy activity, which is important for lifespan extension in these mutants (*Meléndez et al., 2003*; *Guo et al., 2014*; *Lapierre et al., 2013*). Lysosome function is essential for clearance of autophagic substrates. Constitutive autophagy activity leads to more severe defects when lysosome function is compromised (*Sun et al., 2011*). Overexpression of XBP-1s, the activated form of the UPR[ER] transcription factor, is found to increase lysosome activity to promote clearance of toxic proteins and extend *C. elegans* lifespan (*Imanikia et al., 2019*). However, it is unclear whether and how lysosome activity is modulated by longevity-promoting pathways, or how lysosomes contribute to protein homeostasis and lifespan extension.

In this study, we employed cell biology assays to examine lysosomal changes with age in *C. elegans*. We found that various lysosomal properties are altered, which indicates that lysosome activity declines with age. The age-associated lysosomal changes are suppressed in multiple different long-

lived mutant worms, which exhibit increased expression of lysosome genes. Our data suggest that lysosome activity is modulated by longevity pathways and is essential for lifespan extension.

## Results

### Lysosomes undergo age-associated alternations in *C. elegans*

We examined lysosome morphology using the NUC-1::CHERRY reporter in *C. elegans* adults at different ages. Lysosomes appeared mainly as small puncta at day 1 of adulthood in hypodermis, while short tubules were observed at day 3 (*Figure 1A,B*). The tubular lysosomal structures were increased in both length and abundance at day 5, leading to formation of an extensive tubular network at day 9 (*Figure 1C,D,I*). The tubular lysosomal network was still observed in the hypodermis at day 15 of adulthood, indicating that it persisted during aging (*Figure 1—figure supplement 1A*). In aged adults, the number of vesicular lysosomes reduced gradually but the mean volume of each lysosome increased, while the total volume of lysosomes also increased significantly (*Figure 1J–L*). Similar changes in lysosome morphology, number and volume were also observed in body wall muscle cells and intestinal cells with age even though tubular lysosomal structures were less abundant in these two tissues compared to hypodermis (*Figure 1—figure supplement 1C–H*).

We next examined whether other lysosomal properties, including dynamics, acidification and degradation activity, are altered in worms with increased age. To examine lysosome dynamics, we measured Pearson's correlation coefficient to compare the colocalization of lysosomes in two time-lapse image frames taken 60 s apart. We found a higher level of colocalization in adult hypodermis, resulting in a higher Pearson's correlation coefficient than in larvae (*Figure 1M,N*). This suggests that lysosomes are less dynamic in adults. Consistent with this, the velocity of lysosomes was higher in larvae than in adults (*Figure 1O*). The Pearson's correlation coefficient did not change obviously in adults from day 1 to day 9, but the velocity of lysosomes was significantly reduced at days 5 and 9, which suggests that lysosome motility declines with age (*Figure 1N,O*). We examined lysosome acidity by co-staining with LysoTracker Red (LTR) and LysoSensor Green DND-189 (LSG, pKa 5.2) (*Baxi et al., 2017*). LTR is less sensitive to increased acidity than LSG and is used as a control for normalizing the dye intake (*Duvvuri et al., 2004*). The fluorescence intensity ratio of LSG vs LTR (LSG/LTR) is quantified to indicate lysosome acidity. We found that the LSG/LTR ratio in the intestine was reduced in adults at days 3, 5 and 9 compared to day 1, which suggests that lysosome acidity declines in aging adults (*Figure 2A–D'', I*). The tubular lysosomal structures enriched in the hypodermis of aged adults were weakly stained by LysoTracker Red but were not labeled by LysoSensor Green, which suggests that lysosomal tubules may be less acidic than the vesicular ones (*Figure 2—figure supplement 1A–D*). Cathepsin L (CPL-1) is synthesized as an inactive pro-enzyme, which is converted to the active mature form in lysosomes through proteolytic removal of the pro-domain (*Stoka et al., 2016*). The processing of endogenous CPL-1 can be examined by western blot and quantified to indicate the degradation activity of lysosomes. We found that CPL-1 processing reduced significantly in adults at days 5 and 9 compared to day 1, and pro-CPL-1 accumulated with age (*Figure 2N,O*). These results suggest that lysosomal degradation activity decreases with age. Altogether, these data suggest that lysosomes undergo a series of age-associated changes including reduced vesicular but increased tubular morphology, increased mean and total volume, and decreased acidity, motility and degradation activity.

### Lysosome morphology and activity are well maintained in *daf-2* mutants with age

We investigated whether these age-associated lysosomal changes are altered by longevity regulatory factors. Insulin/IGF-1 signaling (IIS) is an evolutionarily conserved aging regulatory pathway. Mutations in the insulin/IGF-1 receptor DAF-2 double the lifespan of wild type (*Kenyon et al., 1993*). We found that lysosomes in the *daf-2(e1370ts)* mutant, which has reduced function of DAF-2, appeared as small puncta and short tubules, and they were not obviously changed with age in hypodermis (*Figure 1E–H* and *Figure 1—figure supplement 1B*). *daf-2(e1370ts)* worms contained significantly more vesicular lysosomes than wild type, and these vesicular lysosomes were smaller in size (*Figure 1J,K*). The tubular lysosomes were shorter in length and they did not form a tubular network in aged *daf-2* adults (*Figure 1I*). The mean and total volume of lysosomes exhibited an age-

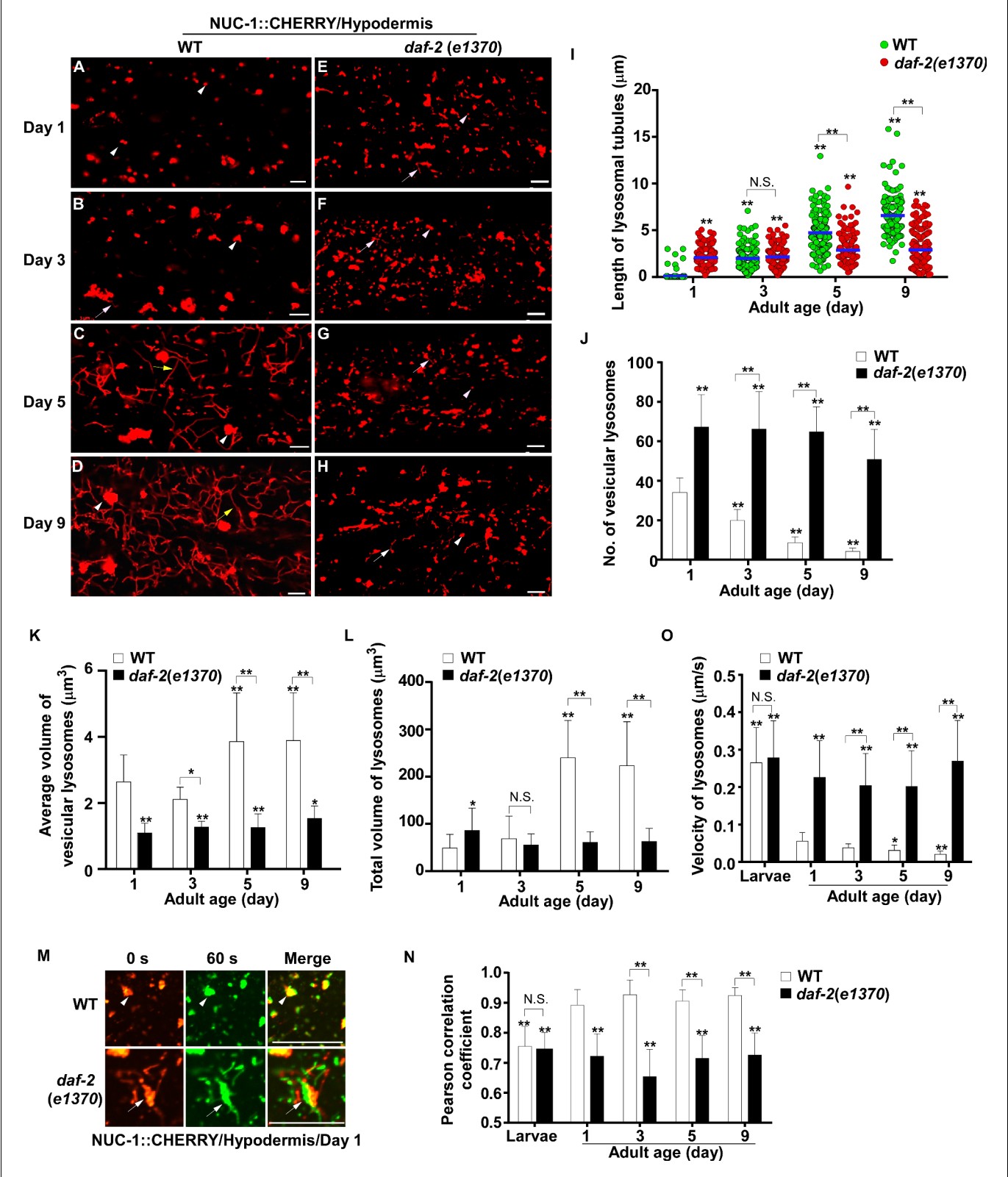

**Figure 1.** Lysosomes exhibit age-associated changes that are suppressed in the long-lived mutant *daf-2*. (A–H) Confocal fluorescence images of the hypodermis in wild type (WT; A–D) and *daf-2(e1370)* (E–H) expressing NUC-1::CHERRY at different ages (adult days 1, 3, 5, 9). White arrowheads indicate vesicular lysosomes; white and yellow arrows indicate short and long lysosomal tubules, respectively. (I–L) Tubule length (I), number (J) and volume (K, L) of lysosomes were quantified in wild type (WT) and *daf-2(e1370)* at different ages. At least 20 (I, J) or 10 (K, L) animals were scored in each

*Figure 1 continued on next page*

*Figure 1 continued*

strain at each day. (M) Time-lapse images of lysosomes in the hypodermis in wild type (WT) and *daf-2(e1370)* expressing NUC-1::CHERRY at adult day 1, with time point 0 s in red and 60 s in green. The overlay (merge) shows lysosome movement over time. Pearson's correlation coefficient and average velocity of lysosomes were determined at the indicated stages and are shown in (N, O). At least 10 animals were scored in each strain at each stage. In (I, J, K, L, N, O), data are shown as mean ± SD. One-way ANOVA with Tukey's multiple comparison test (I) and two-way ANOVA with Fisher's LSD test (J, K, L, N, O) was performed to compare all other datasets with wild type at day 1, or datasets that are linked by lines. *$p<0.05$; **$p<0.001$. All other points had $p>0.05$. N.S., no significance. Scale bars: 5 μm.

The online version of this article includes the following source data and figure supplement(s) for figure 1:

**Source data 1.** Numerical data that are represented as a bar graph in *Figure 1I–L,N,O*.
**Figure supplement 1.** Lysosomes exhibit age-associated alterations in *C. elegans*.
**Figure supplement 1—source data 1.** Numerical data that are represented as a bar graph in *Figure 1—figure supplement 1E–H and J*.

dependent increase in wild type but remained unchanged in *daf-2* adults from day 1 to day 9 (*Figure 1K,L*). Increased number and reduced mean volume of vesicular lysosomes were also observed in body wall muscle cells of *daf-2* mutants at different ages (*Figure 1—figure supplement 1C,E,F*). In the intestine of *daf-2* mutants, the number and mean volume of vesicular lysosomes was similar to that in wild type (*Figure 1—figure supplement 1D,G,H*).

We found that lysosomes in *daf-2* adults at different ages were more dynamic than in wild type. The Pearson's correlation coefficient was lower, and the velocity of lysosomes was higher in *daf-2* adults (*Figure 1M–O*). The lysosome velocity in *daf-2* adults was similar to that in wild-type larvae. The fluorescence intensity ratio of LSG/LTR at days 5 and 9 in *daf-2* was higher than in wild type, and it was similar to the LSG/LTR ratio in wild type at day 1 (*Figure 2E–I*). This suggests that lysosomal acidity is maintained in *daf-2* worms with age. To further examine this, we fused the pH-sensitive fluorescent protein pHTomato with NUC-1, and transiently expressed NUC-1::pHTomato using the heat-shock promoter. At 24 hr post heat-shock treatment, NUC-1::pHTomato overlapped well with the lysosomal membrane protein SCAV-3, indicating delivery of the fusion protein to lysosomes (*Figure 2—figure supplement 1E–G''*). pHTomato has a p*K*a close to 7.8 and thus exhibits increased fluorescence when the pH is increased (*Li and Tsien, 2012*). The average fluorescence intensity of NUC-1::pHTomato in each lysosome was quantified in the hypodermis. Loss of the lysosomal Ca²⁺ channel CUP-5 affects lysosome activity and acidity and causes increased pHTomato intensity in lysosomes (*Figure 2J,L,M*; *Hersh et al., 2002*; *Treusch et al., 2004*; *Sun et al., 2011*; *Miao et al., 2020*). We found that NUC-1::pHTomato intensity was significantly lower in *daf-2 (e1370ts)* mutants than in wild type (*Figure 2J,K,M*). By contrast, the average intensity of NUC-1::CHERRY, which is insensitive to pH, was unchanged in *daf-2* or *cup-5* lysosomes compared with wild type (*Figure 2—figure supplement 1H–K*). Collectively, these data suggest that lysosome acidity increases in *daf-2* worms. We next examined CPL-1 processing and found that significantly more mature CPL-1 was produced in *daf-2* worms than in wild type at different ages, which suggests that the degradation activity of lysosomes is increased in *daf-2* (*Figure 2N,O*). To corroborate this, we examined lysosomal degradation activity using the NUC-1::CHERRY fusion protein. When delivered to lysosomes, CHERRY is cleaved from the fusion protein by cathepsins, and the extent of cleavage can be visualized by Western blot and quantified to indicate the degradation activity of lysosomes (*Miao et al., 2020*). Consistent with the CPL-1 processing assay, we found significantly increased CHERRY cleavage in *daf-2* worms compared to wild type (*Figure 1—figure supplement 1I,J*).

The above results suggest that the properties of lysosomes – including morphology, dynamics, acidity and degradation activity – are well maintained in *daf-2* mutants, but not in wild type, with age. To further test this, we examined lysosomes by high voltage electron microscopy (HVEM). At day 1 of adulthood, 40% of wild-type lysosomes in hypodermis appeared as membrane-enclosed, dense and spherical vesicles (*Figure 3A,B,K*). In addition, around 50% of wild-type lysosomes contained both electron-dense and -lucent contents, with half of them extending electron-lucent tubules (*Figure 3D,E,K*). In addition to the above two main classes, a few lysosomal tubules with either dense (1.4%) or lucent (7.1%) contents were observed (*Figure 3C,F,K*). We found that the ultrastructure of lysosomes changed dramatically in wild type at day 5. The proportion of dense vesicular lysosomes reduced sharply from 40% to 3.7% (*Figure 3K*). Lysosomes with both dense and lucent contents decreased markedly, and they did not extend tubules (*Figure 3K*). Instead, the majority of hypodermal lysosomes at day 5 (81.4%) were lucent tubules that formed a tubular network, which is

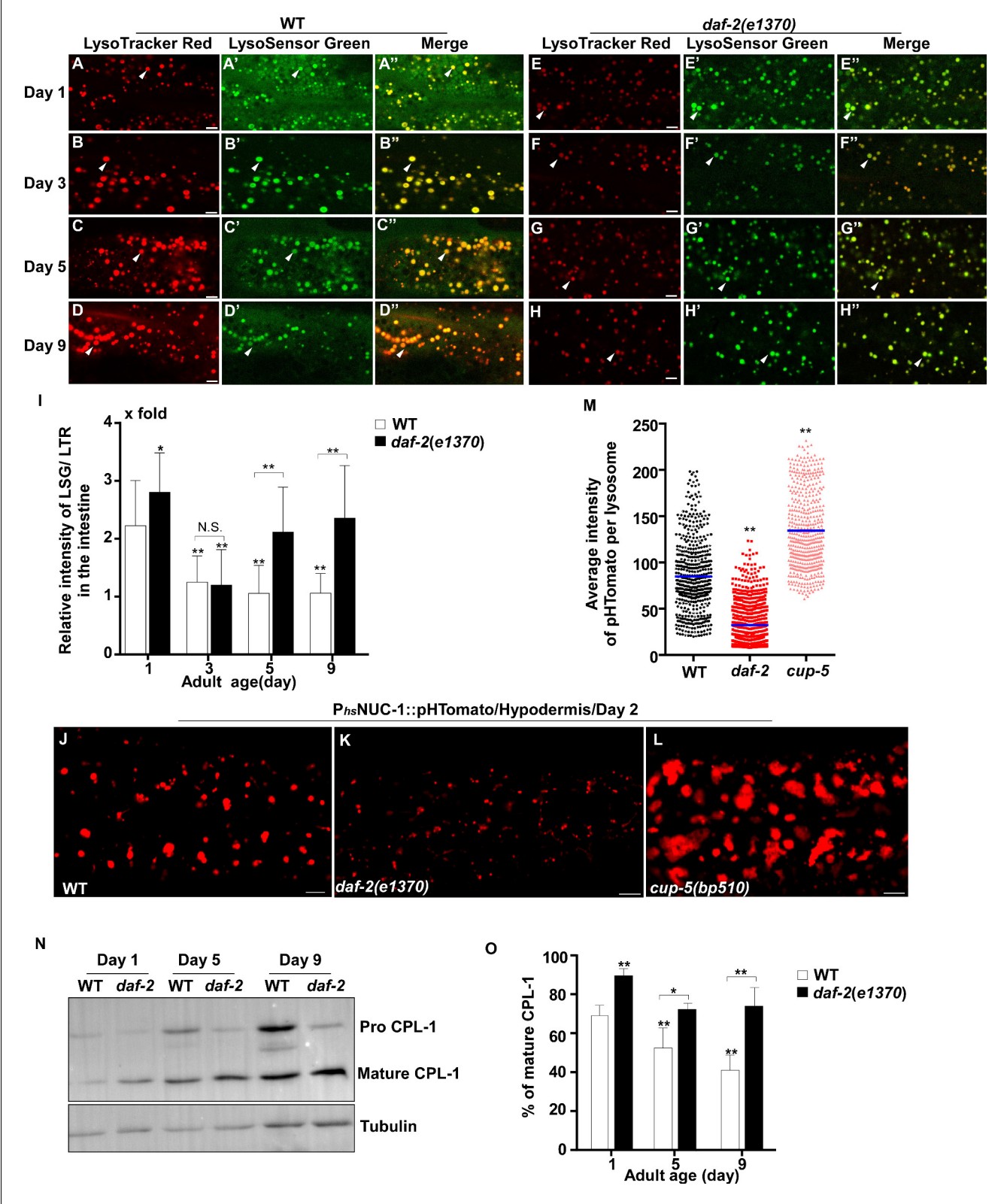

**Figure 2.** Lysosomal acidity and degradation activity are increased in *daf-2*. (A–H") Confocal fluorescence images of the intestine in wild type (WT; A–D") and *daf-2(e1370)* (E–H") adults at different ages stained by LSG DND-189 and LTR DND-99. (I) The relative intensity of LSG/LTR in wild type and *daf-2(e1370)* at different ages was quantified. At least 10 animals were scored in each strain at each day. (J–L) Confocal fluorescence images of the hypodermis at adult day 2 in wild type (WT; J), *daf-2(e1370)* (K) and *cup-5(bp510)* (L) expressing NUC-1::pHTomato controlled by the heat-shock (hs)

*Figure 2 continued on next page*

*Figure 2 continued*

promoter. The average intensity of pHTomato per lysosome is shown in (M). At least 20 animals were scored in each strain. (N) Western blot analysis of CPL-1 processing in wild type (WT) and *daf-2(e1370)* at different adult ages. The percentage of mature CPL-1 was quantified (O). Three independent experiments were performed. In (I, M, O), data are shown as mean ± SD. One-way ANOVA with Tukey's multiple comparisons test (I, M) or two-way ANOVA with Fisher's LSD test (O) was performed to compare all other datasets with wild type (M) or wild type at day 1 (I, O), or to compare datasets that are linked by lines. *p<0.05; **p<0.001. All other points had p>0.05. N.S., no significance. Scale bars: 5 µm.

The online version of this article includes the following source data and figure supplement(s) for figure 2:

**Source data 1.** Numerical data that are represented as a bar graph in *Figure 2I,M and O*.

**Figure supplement 1.** pHTomato can be used to probe lysosomal acidity in *C. elegans*.

**Figure supplement 1—source data 1.** Numerical data that are represented as a bar graph in *Figure 2—figure supplement 1C,D and K*.

in good agreement with the observations by fluorescence microscopy (*Figures 1A,C* and *3G,K*). In *daf-2(e1370ts)* mutants, most lysosomes at day 1 (70.8%) appeared as dense spherical vesicles that were significantly smaller than in wild type (*Figure 3H,L,M*). In addition, vesicular lysosomes with both dense and lucent contents were observed (*Figure 3J,L*). Importantly, the ultrastructure of lysosomes did not change obviously in *daf-2* worms at day 5, except for a slightly higher percentage of dense tubules (1.4% vs 7.1%) and a lower percentage of the dense-lucent vesicles (27.8% vs 17.1%; *Figure 3L*). These HVEM data are consistent with the observations by fluorescence microscopy and together they indicate that lysosomal morphology and properties are well maintained in *daf-2* mutants with age.

## Lysosome activity is increased in *eat-2* and *isp-1* mutants

Our data suggest that reducing IIS suppresses age-associated changes in lysosomal shape, size, dynamics, acidity and degradation activity. We next examined whether lysosome patterns and activity are altered in two other long-lived mutants, *eat-2* and *isp-1*, which extend lifespan through restricted caloric intake and impaired mitochondrial respiration, respectively (*Lakowski and Hekimi, 1998*; *Feng et al., 2001*). We found that lysosome patterns in *isp-1(qm150)* and *eat-2(ad1116)* mutants at different ages resembled those in *daf-2(e1370ts)*, except that tubular lysosomal structures were more abundant in *eat-2(ad1116)* than in *daf-2* and *isp-1* worms (*Figure 4A–L*). Like in *daf-2* worms, the number of vesicular lysosomes increased, and the mean volume decreased in *eat-2* and *isp-1* mutants; tubular lysosomes at days 5 and 9 were shorter in length and did not form tubular networks (*Figure 4M–O*). The velocity of lysosomes was significantly higher in *eat-2* worms than in wild type at different ages, while *isp-1* lysosomes had a higher motility than wild type at day 1 and day 9 (*Figure 4—figure supplement 1A*). By examining the fluorescence intensity ratio of LSG/LTR, we found that lysosome acidity was significantly higher in *eat-2(ad1116)* worms than in wild type at all adult ages tested, while increased lysosome acidity was seen in *isp-1(qm150)* mutants at days 3 and 5 but not day 9 (*Figure 4—figure supplement 1B–J*). In agreement with this, the average intensity of NUC-1::pHTomato was significantly lower in *eat-2* and *isp-1* than in wild type, which suggests that lysosome acidity was increased (*Figure 4P–S*). We found that more mature CPL-1 was produced in *eat-2(ad1116)* and *isp-1(qm150)* mutants than in wild type at different ages except for *isp-1* at day 9, where the percentage of mature CPL-1 was similar to wild type (*Figure 4T–W*). Collectively, these data suggest that like the IIS mutant *daf-2*, lysosome morphology, motility, acidity and degradation activity are well maintained with age in *eat-2* mutants, whereas the appearance of age-related lysosomal changes is delayed in *isp-1(qm150)* worms.

## Expression of lysosome-related genes increases in long-lived worms in a DAF-16- and SKN-1-dependent manner

To investigate how lysosome activity is maintained in long-lived worms, we examined the expression of 85 lysosome-related genes by quantitative PCR (qRT-PCR). These genes encode lysosomal membrane proteins, hydrolases and components of the proton pump V-ATPase (*Figure 5A* and *Supplementary file 1*). We found that expression of 43 lysosomal genes was significantly reduced at day 5 compared to day 1. They included 15 *vha* genes encoding subunits of the V-ATPase and 17 cathepsin genes encoding lysosomal proteases, which is consistent with reduced lysosomal acidity and degradation activity with age (*Figure 5A–C* and *Supplementary file 1*, *2*). In addition, 13

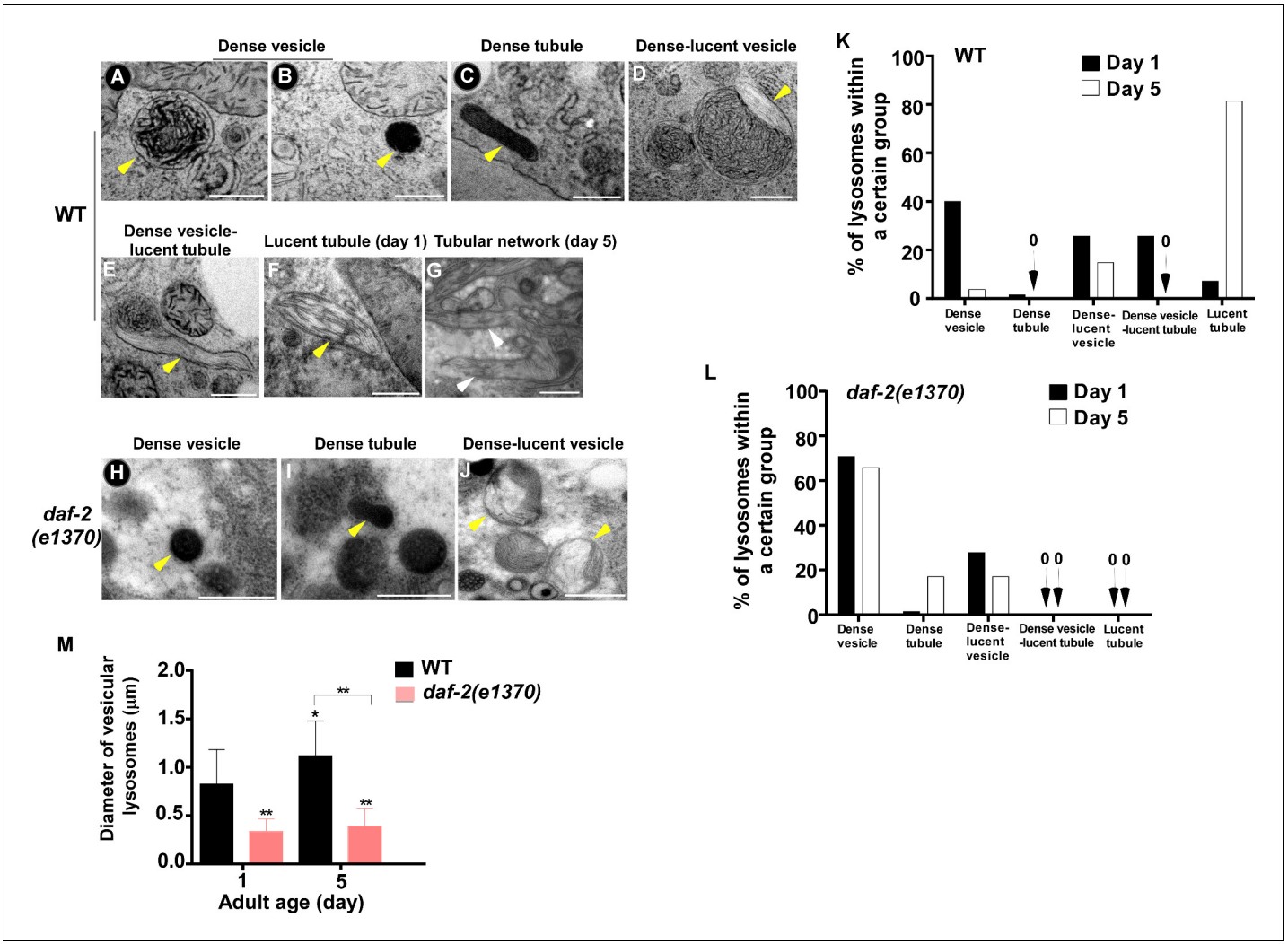

**Figure 3.** The ultrastructure of lysosomes changes dramatically in wild type with age but is maintained well in *daf-2*. (A–J) Representative HVEM images of lysosomes in the hypodermis in wild type (WT; A–G) and *daf-2 (e1370)* (H–J). Yellow arrowheads indicate vesicular lysosomes or short lysosomal tubules. White arrows indicate the lysosomal tubular network formed in wild type at day 5. Scale bars: 500 nm. (K, L) The percentage of lysosomes within a certain morphology group revealed by HVEM was quantified in wild type (WT; K) and *daf-2(e1370)* (L) at different ages (day 1 and day 5). At least 70 lysosomes were quantified in each strain at each age. (M) The diameter of vesicular lysosomes in wild type (WT) and *daf-2(e1370)* at different ages was quantified. At least 50 vesicular lysosomes were counted in each strain at each age. One-way ANOVA with Tukey's multiple comparisons test was performed to compare all other datasets with wild type at day 1, or datasets that are linked by lines. *p<0.05; **p<0.001.

The online version of this article includes the following source data for figure 3:

**Source data 1.** Numerical data that are represented as a bar graph in *Figure 3K–M*.

lysosomal genes exhibited increased expression with age and the expression of 29 lysosomal genes was unaltered at day 5 compared to day 1 (*Figure 5A* and *Supplementary file 3*, *4*).

Among the 43 lysosome genes whose expression declined with age, 20 exhibited significantly increased expression in *daf-2* mutants compared to wild type at adult day 1 (*Figure 5D* and *Supplementary file 5*). These 20 genes mainly encode lysosomal hydrolases, including eight cathepsin proteases and six hydrolases that digest carbohydrates and lipids (*Figure 5D* and *Supplementary file 5*). In *isp-1(qm150)* mutants, 10 out of the 43 lysosomal genes were upregulated (*Figure 5E* and *Supplementary file 5*). Nine of the upregulated genes encode hydrolases and expression of all of them, except for *cpr-8*, is increased in *daf-2* mutants (*Figure 5D,E*, *Figure 6— figure supplement 1A* and *Supplementary file 5*). In *eat-2(ad1116)* mutants, 14 out of the 43 lysosome genes were upregulated and 8 of them encode V-ATPase subunits (*Figure 5F* and

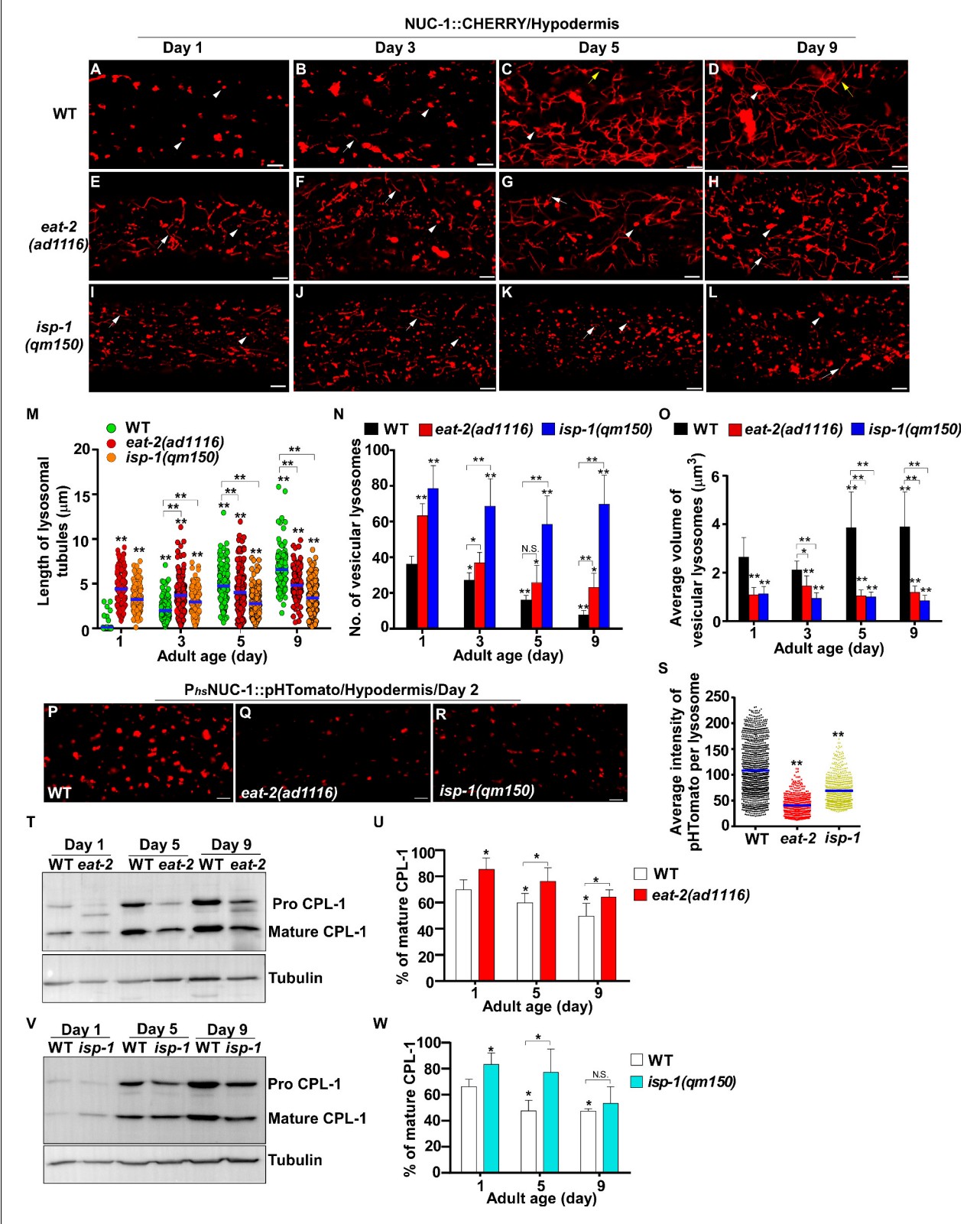

**Figure 4.** *eat-2(ad1116)* and *isp-1(qm150)* mutants exhibit increased lysosome activity. (**A–L**) Confocal fluorescence images of the hypodermis in wild type (WT; **A–D**), *eat-2(ad1116)* (**E–H**) and *isp-1(qm150)* (**I–L**) expressing NUC-1::CHERRY at different adult ages. White arrowheads indicate vesicular lysosomes; white and yellow arrows designate short and long lysosomal tubules, respectively. (**M–O**) The length of tubular lysosomes (**M**), and the number (**N**) and mean volume (**O**) of vesicular lysosomes were quantified in wild type (WT), *eat-2(ad1116)* and *isp-1 (qm150)* at different ages. At least

*Figure 4 continued*

10 animals were scored in each strain at each age. (P–R) Confocal fluorescence images of the hypodermis in wild type (WT; **P**), *eat-2(ad1116)* (**Q**) and *isp-1(qm150)* (**R**) expressing NUC-1::pHTomato controlled by the heat-shock (hs) promoter. The average intensity of pHTomato per lysosome was quantified (**S**). At least 20 animals were scored in each strain. (**T, V**) Western blot analysis of CPL-1 processing in *eat-2(ad1116)* (**T**) and *isp-1(qm150)* (**V**) at different ages. The percentage of mature CPL-1 was quantified (**U, W**). Three independent experiments were performed. In (**M, N, O, S, U, W**), data are shown as mean ± SD. One-way ANOVA with Tukey's multiple comparisons test (**M, S**) or two-way ANOVA with Fisher's LSD test (**N, O, U, W**) was performed to compare all other datasets with wild type (**S**) or wild type at day 1 (**M, N, O, U, W**) or datasets that are linked by lines. *p<0.05; **p<0.001. All other points had p>0.05. N.S., no significance. Scale bars: 5 μm.

The online version of this article includes the following source data and figure supplement(s) for figure 4:

**Source data 1.** Numerical data that are represented as a graph in *Figure 4M,N,O,S,U and W*.

**Figure supplement 1.** Lysosome acidity and motility increase in *eat-2(ad1116)* and *isp-1(qm150)* mutants.

**Figure supplement 1—source data 1.** Numerical data that are represented as a graph in *Figure 4—figure supplement 1A and J*.

*Supplementary file 5*). By contrast, the expression of very few *vha* genes was increased in *daf-2* and *isp-1* mutants (*Figure 5D,E* and *Figure 6—figure supplement 1A*).

We next examined transcription factors that act downstream of the three longevity pathways.

The transcription factors DAF-16/FOXO, SKN-1/NRF2 and HSF-1 all respond to reduced IIS. We found that loss of *daf-16* and *skn-1* led to reduced expression of 13 and 8 lysosomal genes, respectively, in *daf-2* worms (*Figure 6A,B*). We examined the six lysosomal genes whose expression was reduced by both *daf-16* and *skn-1* mutations (*Figure 6—figure supplement 1B*). Expression of these lysosomal genes did not further reduce in *daf-16;daf-2;skn-1* triple mutants, which suggests that DAF-16 and SKN-1 act in the same genetic pathway to regulate their expression (*Figure 6—figure supplement 1B*). Consistent with this, loss of *daf-16* or *skn-1* led to increased pHTomato intensity, reduced fluorescence intensity ratio of LSG/LTR and reduced CHERRY cleavage in *daf-2* mutants, which indicates that DAF-16 and SKN-1 function is important for elevation of lysosome acidity and degradation activity in *daf-2* mutants (*Figures 6C–G,I,K–N* and *Figure 6—figure supplement 1C–H''*). The pHTomato intensity, LSG/LTR fluorescence intensity ratio and CHERRY cleavage were not further altered in *daf-16;daf-2;skn-1* triple mutants, consistent with co-regulation of lysosomal gene expression by DAF-16 and SKN-1 when IIS is impaired (*Figure 6H–N* and *Figure 6—figure supplement 1I–J''*). HLH-30 is the putative *C. elegans* homolog of human TFEB, a master transcription factor for autophagy and lysosome biogenesis (*O'Rourke and Ruvkun, 2013*, *Settembre et al., 2011*). It was reported recently that both HLH-30 and DAF-16 are required for the longevity of *daf-2* mutants and they act as combinatorial transcription factors to fulfill this function (*Lin et al., 2018*). We found that loss of *hlh-30* caused reduced expression of 6 hydrolase genes in *daf-2* mutants, and 5 of them were also targeted by DAF-16 (*Figure 6—figure supplement 1K,L*). However, unlike *daf-16(lf)*, loss of *hlh-30* did not affect NUC-1::CHERRY cleavage in *daf-2* worms, which suggests that lysosome degradation activity may be unaltered (*Figure 6—figure supplement 1M*). The CHERRY cleavage in *daf-16;daf-2;hlh-30* was higher than in *daf-16;daf-2*, suggesting that loss of *hlh-30* may have a beneficial effect on lysosomal degradation in *daf-16;daf-2* (*Figure 6—figure supplement 1M*). Loss of *hsf-1* had no effect on lysosomal gene expression or NUC-1::CHERRY cleavage in *daf-2* worms, which suggests that HSF-1 is dispensable for lysosome regulation in *daf-2* mutants (*Figure 6—figure supplement 1N,O*).

In addition to responding to insulin signaling, DAF-16 also acts downstream of the mitochondrial pathway, while *skn-1* RNAi reduces the lifespan of *eat-2* (*Senchuk et al., 2018*; *Park et al., 2010*). We found that loss of *daf-16* and *skn-1* also affected lysosome gene expression in *eat-2* and *isp-1* mutants (*Figures 7A,B* and *8A,B*). Loss of either *daf-16* or *skn-1* caused reduced expression of *vha-12* and *vha-15* in *eat-2* mutants, which was further decreased in *daf-16;eat-2;skn-1* triple mutants (*Figure 7—figure supplement 1A*). These results indicate that DAF-16 and SKN-1 have additive effects on the expression of *vha-12* and *vha-15*. Consistent with this, the NUC-1::pHTomato intensity in lysosomes was higher and the LSG/LTR fluorescence intensity ratio was lower in *daf-16;eat-2* and *eat-2;skn-1* than in *eat-2*, and these parameters were further altered in *daf-16;eat-2;skn-1* (*Figure 7C–L*). In addition, cleavage of CHERRY from NUC-1::CHERRY was reduced in *daf-16;eat-2* and *eat-2;skn-1* compared to *eat-2* single mutants, which suggests that lysosomal degradation activity is also affected (*Figure 7—figure supplement 1C,D*). However, CHERRY cleavage was not further decreased in *daf-16;eat-2;skn-1* (*Figure 7—figure supplement 1C,D*). In *isp-1* mutants, expression

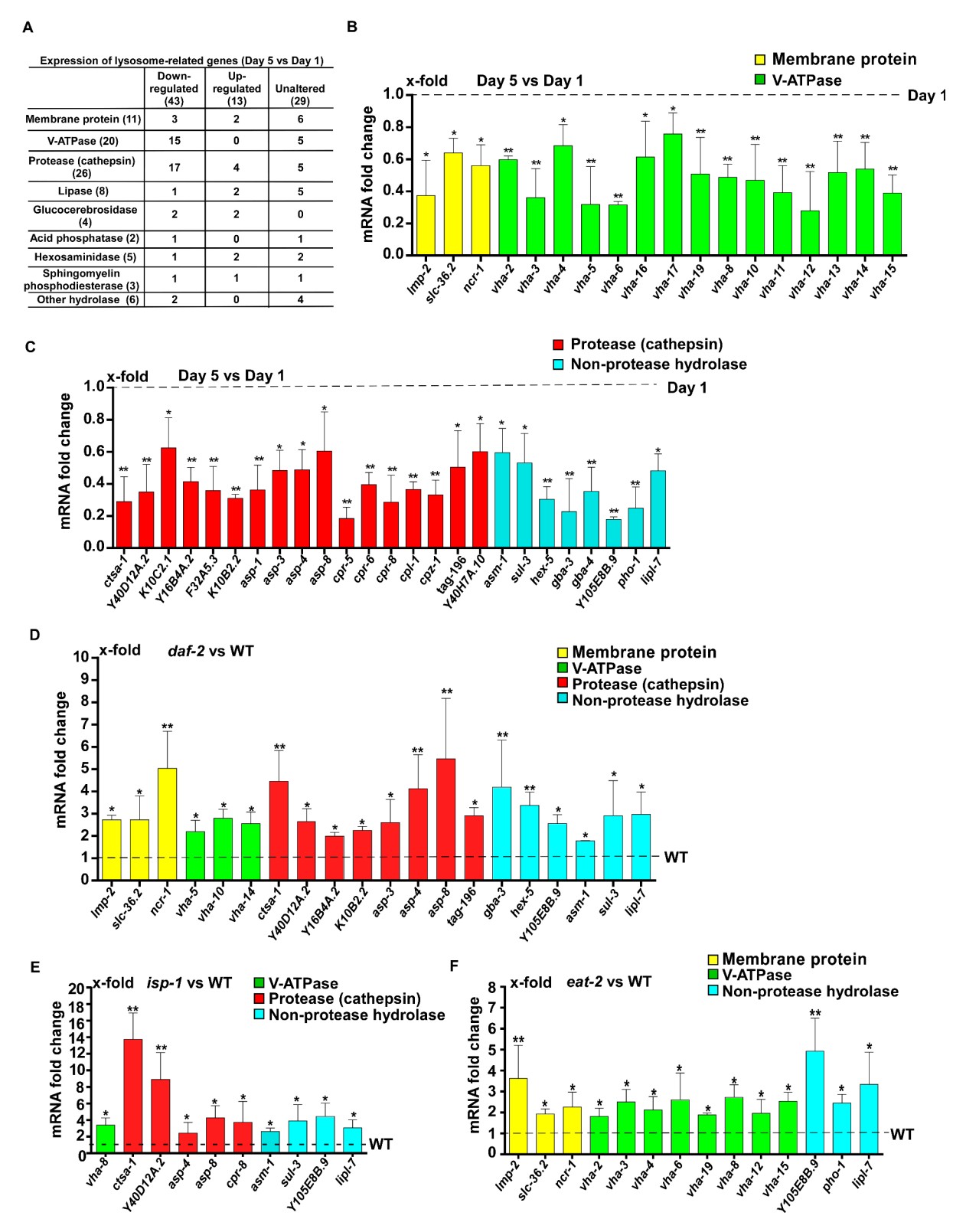

**Figure 5.** Lysosomal gene expression is upregulated in the long-lived mutants *daf-2*, *eat-2* and *isp-1*. (A) Expression of 85 lysosome-related genes in wild type at day 1 and day 5 was analyzed. 43 and 13 lysosomal genes were down- and up-regulated with age, respectively. Expression of 29 lysosomal genes was unaltered at day 5 compared with day 1. (B, C) Quantitative RT-PCR (qRT-PCR) analyses of the 43 downregulated lysosomal genes in wild type at day 1 and day 5. (D–F) Expression of the 43 downregulated lysosomal genes was analyzed by qRT-PCR at day 1 in *daf-2* (D), *isp-1* (E) and *eat-2*

*Figure 5 continued on next page*

*Figure 5 continued*

(**F**) worms. In (**B–F**), three independent experiments were performed. The transcription level of lysosomal genes in wild type (WT) at day 1 was normalized to '1' for comparison. Data are shown as mean ± SD. Multiple *t* testing was performed to compare mutant datasets with wild type. *p<0.05; **p<0.001.

The online version of this article includes the following source data for figure 5:

**Source data 1.** Numerical data that are represented as a bar graph in *Figure 5B–F*.

of 4 hydrolase genes (*asp-4, asp-8, asm-1* and *Y105E8B.9*) was affected by both *daf-16* mutation and *skn-1* RNAi, and their expression in triple mutants (*daf-16;isp-1skn-1* RNAi) was similar to the double mutant (*Figure 7—figure supplement 1B*). These results suggest that DAF-16 and SKN-1 act together to regulate lysosomal gene expression in *isp-1*. In agreement with this, loss of *skn-1* or *daf-16* led to increased pHTomato intensity and reduced LSG/LTR fluorescence intensity ratio in *isp-1* lysosomes, while these parameters remained unchanged in *daf-16;isp-1skn-1* RNAi worms (*Figure 8C–L*). The *daf-16* mutation caused reduced NUC-1::CHERRY cleavage in *isp-1*, while *skn-1* RNAi did not obviously affect CHERRY cleavage in *isp-1* or *daf-16;isp-1* (*Figure 7—figure supplement 1E,F*). We found that loss of PHA-4/FOXA, the key downstream effector of the dietary restriction pathway, had no effect on lysosomal gene expression in *eat-2* mutants, while loss of HIF-1, the transcription factor acting downstream of the mitochondrial pathway, did not reduce expression of lysosomal genes in *isp-1* mutants, except for *asp-8* (*Figures 7M* and *8M*). Loss of *pha-4* and *hif-1* had no effect on the acidity and degradation activity of lysosomes in *eat-2* and *isp-1* mutants, respectively (*Figures 7N–R* and *8N–R* and *Figure 7—figure supplement 1G–J*). Altogether, these data suggest that PHA-4 and HIF-1 are dispensable for lysosome regulation in *eat-2* and *isp-1* mutants.

## Lysosome function is important for clearance of aggregate-prone proteins and for lifespan extension induced by multiple mechanisms

Protein insolubility or aggregation is an inherent part of normal aging due to reduced proteostasis with age. We tested whether the decline in lysosome function contributes to the accumulation of protein aggregates. NMY-2 was previously identified as an aggregation-prone protein which becomes more insoluble with age (*David et al., 2010*; *Bohnert and Kenyon, 2017*). Consistent with this, NMY-2::GFP fluorescence was almost invisible in wild-type oocytes at day 1 of adulthood, but was visible as GFP puncta at day 5 (*Figure 9A,B* and *Figure 9—figure supplement 1A,B*). Loss of CUP-5, the lysosomal $Ca^{2+}$ channel homologous to human TRPML, caused increased pHTomato intensity in lysosomes and reduced fluorescence intensity ratio of LSG/LTR, which indicates that lysosomal acidity is affected (*Figure 2L,M* and *Figure 9—figure supplement 1P–X*). Moreover, CPL-1 processing was reduced significantly in *cup-5* mutants at all adult ages tested, which is suggestive of defects in lysosomal degradation activity (*Figure 9—figure supplement 1Y,Z*). In *cup-5*, NMY-2::GFP fluorescence increased significantly in oocytes at days 1 and 5, but the number of NMY-2::GFP puncta was not obviously increased (*Figure 9C,D,I–K* and *Figure 9—figure supplement 1C,D,M–O*). The number of NMY-2::GFP puncta was reduced significantly in oocytes of *daf-2, eat-2* and *isp-1* mutants at day 5, consistent with decreased formation and/or accumulation of protein aggregates (*Figure 9F,K* and *Figure 9—figure supplement 1F,J,O*). We found that loss of *cup-5* caused significantly increased NMY-2::GFP fluorescence in *daf-2, eat-2* and *isp-1* oocytes at both day 1 and day 5, and the number of NMY-2::GFP puncta also increased at day 5 (*Figure 9G–K* and *Figure 9—figure supplement 1G–O*). This suggests that lysosome function is important for clearance of aggregation-prone proteins and protein aggregates in long-lived worms. We observed that the *cup-5* mutation was more potent at increasing the NMY-2::GFP fluorescence than the number of visible NMY-2::GFP aggregates (*Figure 9A–K* and *Figure 9—figure supplement 1A–O*). This suggests that more soluble or lower-molecular-weight forms of aggregate-prone proteins may be removed more efficiently by lysosomes.

Finally, we examined whether lysosome function contributes to lifespan extension. The lysosome-defective mutants *cup-5(bp510)* and *cpl-1(qx304)* were slightly short-lived compared with wild type, and both of these mutations significantly reduced the lifespan in *daf-2, eat-2* and *isp-1* worms (*Figure 9L–Q*). These data indicate that lysosome function is important for lifespan extension

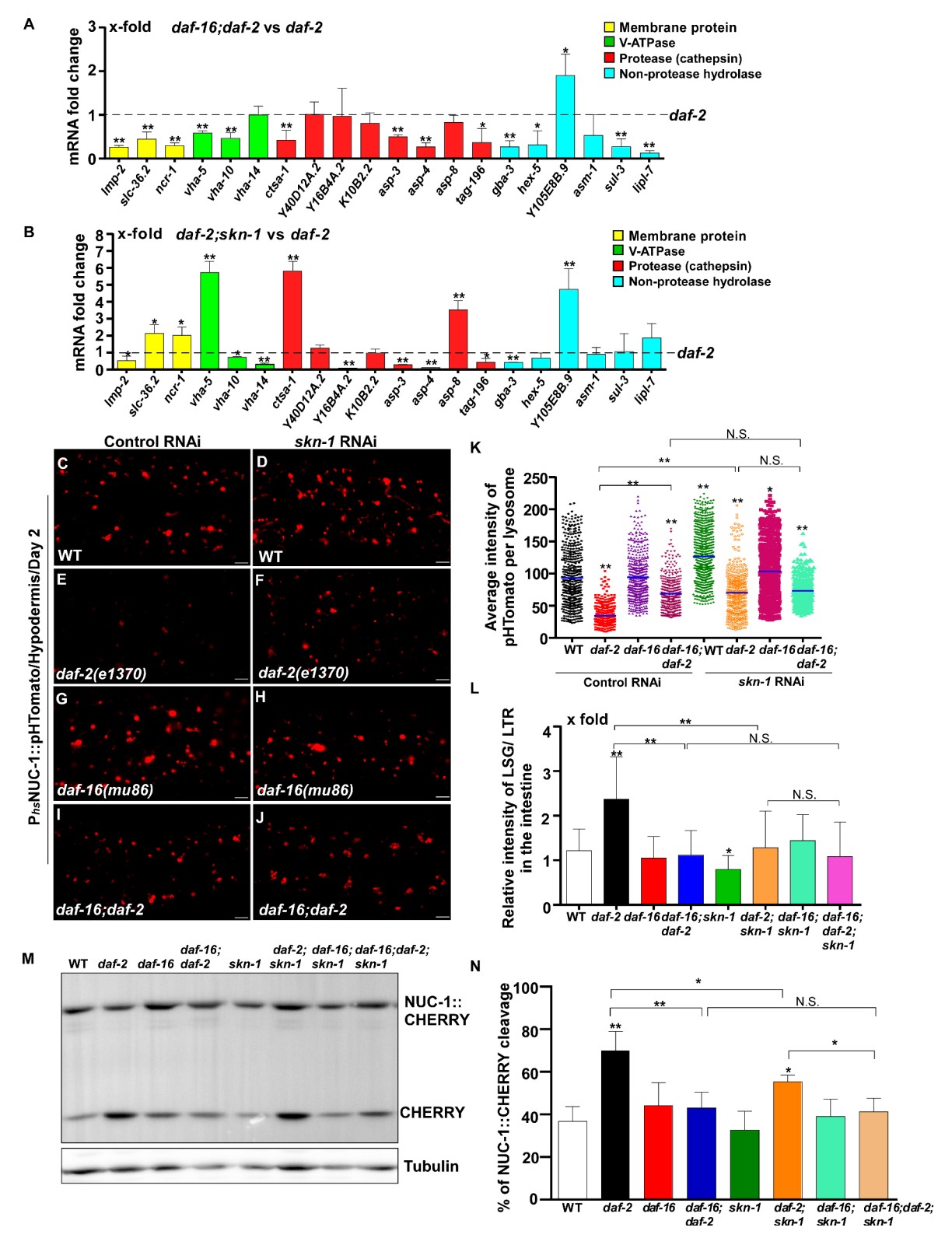

**Figure 6.** DAF-16 and SKN-1 are required for the upregulation of lysosomal genes and maintenance of lysosomal acidity and activity in *daf-2* mutants. (A, B) Expression of the 20 upregulated lysosomal genes in *daf-2* mutants was analyzed by qRT-PCR in *daf-16;daf-2* (A) and *daf-2;skn-1* (B) worms at day 1. Three independent experiments were performed. The transcription level of lysosomal genes in *daf-2(e1370)* at day 1 was normalized to '1' for comparison. (C–J) Confocal fluorescence images of the hypodermis in the indicated strains expressing NUC-1::pHTomato controlled by the heat-shock

*Figure 6 continued on next page*

*Figure 6 continued*

(hs) promoter. Scale bars: 5 μm. The average intensity of pHTomato per lysosome was quantified (K). At least 20 animals were scored in each strain. (L) The relative intensity of LSG/LTR in the intestine was quantified in the indicated strains at day 2. At least 10 animals were scored in each strain. (M) Western blot analysis of CHERRY cleavage from NUC-1::CHERRY in the indicated strains at day 1. Quantification is shown in (N). Three independent experiments were performed. In (A, B, K, L, N), data are shown as mean ± SD. Multiple *t* testing (A, B), or one-way ANOVA with Tukey's multiple comparisons test (K, L, N) was performed to compare datasets of double mutants with *daf-2* (A, B) or to compare all other datasets with wild type (L, N) or with wild type treated with control RNAi (K), or datasets that are linked by lines (K, L, N). *$p < 0.05$; **$p < 0.001$. All other points had $p > 0.05$. N.S., no significance.

The online version of this article includes the following source data and figure supplement(s) for figure 6:

**Source data 1.** Numerical data that are represented as a bar graph in *Figure 6A,B,K,L,N*.
**Figure supplement 1.** DAF-16 and SKN-1 play an overlapping role in regulating lysosomal gene expression in *daf-2* mutants.
**Figure supplement 1—source data 1.** Numerical data that are represented as a bar graph in *Figure 6—figure supplement 1B,L–O*.

induced by multiple mechanisms including reduced IIS, caloric restriction and impaired mitochondrial respiration.

## Discussion

In this study, we investigated how lysosomes change with age and contribute to lifespan regulation. Our data indicate that lysosomes undergo a series of age-associated alterations in *C. elegans* including shape, size, motility, acidity and degradation activity, which suggest a decline in lysosomal function with age. We found that lysosomes are modulated by multiple longevity regulatory pathways, and lysosome function is essential for lifespan extension.

### Various lysosomal properties are altered with age

Age-related increases in the number and size of lysosomes have been observed previously in several species such as *Paramecium*, nematodes and human cell lines (*Sundararaman and Cummings, 1976*; *Epstein, 1972*; *Lipetz and Cristofalo, 1972*; *Brandes et al., 1972*). By employing cell biology assays, we found that lysosomes undergo a series of age-related changes including increased mean and total volume, and decreased motility, acidity and degradation activity. This indicates that the overall function of lysosomes declines with age, which explains in part the age-dependent decline in protein degradation described in various systems (*Cuervo and Dice, 1998*). We observed that lysosomal morphology changes dramatically with age, manifested as greatly increased tubular morphology and a concomitant decrease in vesicular lysosomes. Tubular structures have been observed in the lysosome reformation process when lysosomal contents are retrieved from phagolysosomes or autolysosomes (*Yu et al., 2010*; *Gan et al., 2019*). Moreover, stimulation of macrophages and dendritic cells (DCs) with agonists including LPS leads to reorganization of lysosomes into a tubular network (*Hipolito et al., 2018*). These lysosomal tubules may be induced to fulfil a variety of functions, such as expanding lysosomal volume, promoting phagosome maturation, cargo sorting and exchange, and helping delivery of peptide-loaded MHC-II molecules to the cell surface (*Hipolito et al., 2018*; *Hipolito et al., 2019*; *Mantegazza et al., 2014*; *Boes et al., 2002*; *Boes et al., 2003*; *Chow et al., 2002*; *Vyas et al., 2007*). In *C. elegans*, we found previously that catalytically active lysosomal tubules are formed during molting to promote cuticle replacement (*Miao et al., 2020*). In aged adults, however, lysosomal tubules are static and are not readily stained by LysoSensor Green (*Figure 1N,O* and *Figure 2—figure supplement 1B–D*). Lysosome degradation activity, indicated by CPL-1 processing, is obviously reduced in aged adults. Thus, the lysosomal tubules enriched in aged adults are probably catalytically inactive. The HVEM analyses revealed that young adult worms contain electron-lucent tubules emanating from electron-dense granules, consistent with retrieval and/or recycling of lysosomal contents through tubules. In aged worms, the vast majority of lysosomes are seen as electron-lucent tubules that form a tubular network, whereas very few dense vesicular lysosomes are present (*Figure 3G,K*). It is possible that the lysosomal retrieval, cargo sorting and/or catabolite recycling processes occur inefficiently in aged adults, which leads to accumulation of catalytically inactive tubular lysosomal structures. Future studies are needed to understand how lysosomal tubules are formed in aging adults and whether and how they alter degradation, retrieval or recycling of lysosomal contents.

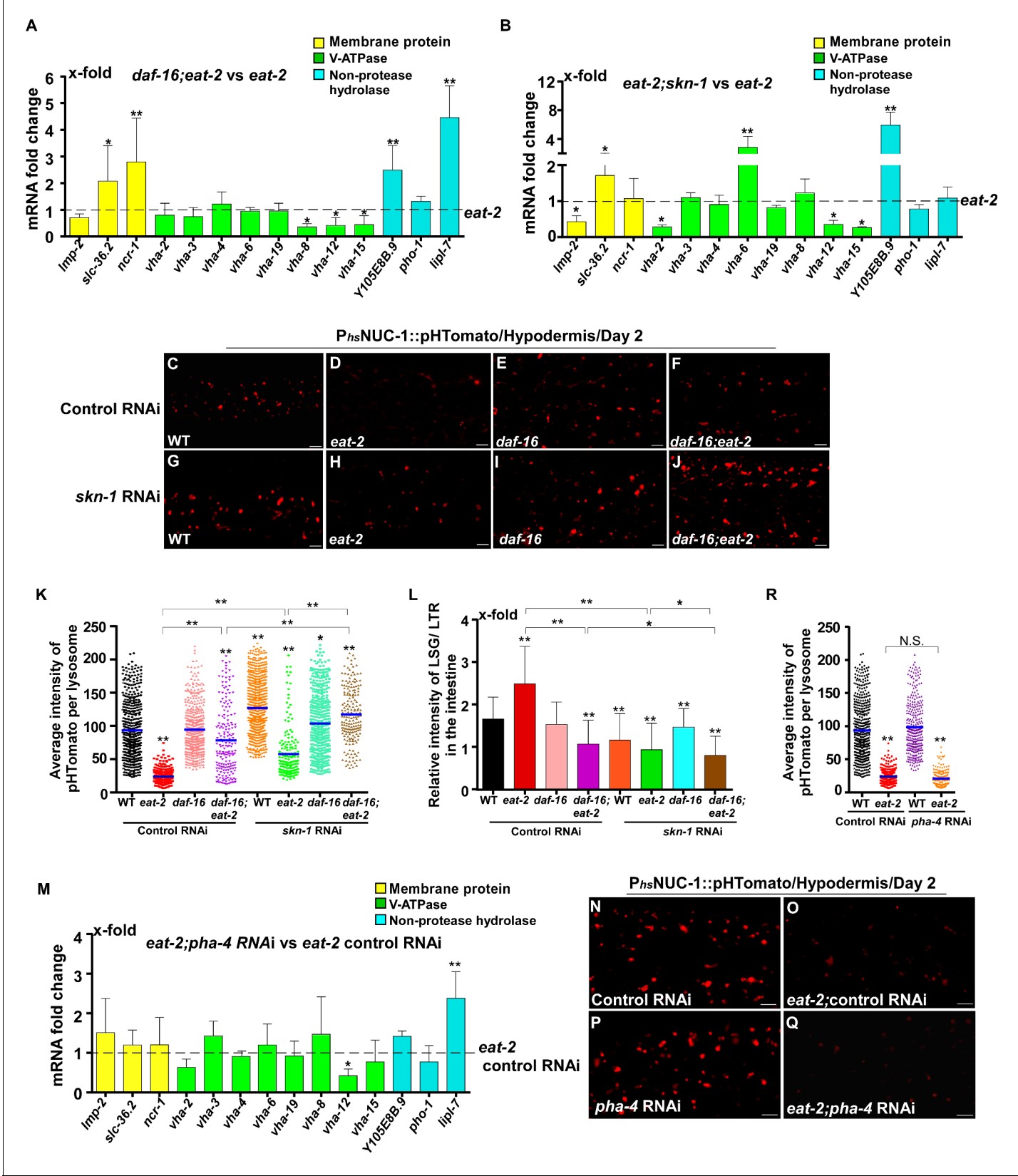

**Figure 7.** DAF-16 and SKN-1, but not PHA-4, regulate lysosomal acidity and gene expression in *eat-2* mutants. (**A, B, M**) Expression of the 14 upregulated lysosomal genes in *eat-2(ad1116)* was analyzed by qRT-PCR in *daf-16;eat-2* (**A**), *eat-2;skn-1* (**B**) and *eat-2;pha-4* RNAi (**M**) worms at day 1.

*Figure 7 continued on next page*

*Figure 7 continued*

Three independent experiments were performed. The transcription level of lysosomal genes in *eat-2(ad1116)* (**A**, **B**) or *eat-2(ad1116)* control RNAi (**M**) at day 1 was normalized to '1' for comparison. (**C–J**, **N–Q**) Confocal fluorescence images of the hypodermis in the indicated strains expressing NUC-1::pHTomato controlled by the heat-shock (hs) promoter. Scale bars: 5 µm. The average intensity of pHTomato per lysosome was quantified (**K**, **R**). At least 20 animals were scored in each strain. (**L**) The relative intensity of LSG/LTR in the intestine was quantified in the indicated strains at day 2. At least 10 animals were scored in each strain. In (**A**, **B**, **K**, **L**, **M**, **R**), data are shown as mean ± SD. Multiple *t* testing (**A**, **B**, **M**) or one-way ANOVA with Tukey's multiple comparisons test (**K**, **L**, **R**) was performed to compare datasets of double mutants with *eat-2* (**A**, **B**), or *eat-2* control RNAi (**M**), or to compare all other datasets with wild type treated with control RNAi (**K**, **L**, **R**), or datasets that are linked by lines (**K**, **L**, **R**). *p<0.05; **p<0.001. All other points had p>0.05. N.S., no significance.

The online version of this article includes the following source data and figure supplement(s) for figure 7:

**Source data 1.** Numerical data that are represented as a bar graph in *Figure 7A,B,K–M,R*.

**Figure supplement 1.** DAF-16 and SKN-1 regulate lysosomal gene expression in *eat-2* and *isp-1* mutants in different manners.

**Figure supplement 1—source data 1.** Numerical data that are represented as a bar graph in *Figure 7—figure supplement 1A,B,D,F,H,J*.

Consistent with changes in multiple lysosomal properties, we observed an age-related decline in the expression of 43 lysosome-related genes (*Figure 5A–C* and *Supplementary file 2*). This affects two main classes of lysosomal proteins, the cathepsin proteases (17 genes) and subunits of the proton pump V-ATPase (15 genes), which may account for the age-associated decline in lysosomal acidity and degradation. We observed that *cpl-1* gene expression declines, but the total CPL-1 protein level appears to increase with age in wild type. The increase in the total CPL-1 protein level is probably caused by reduced CPL-1 processing (*Figures 2N* and *4T–W*) and a decline in CPL-1 protein turnover, consistent with the decline in lysosome activity in aging worms. In addition to decreased expression of 43 lysosome genes, 13 lysosome genes exhibit increased expression with age (*Figure 5A* and *Supplementary file 3*). This may reflect a feed-back response caused by reduced lysosomal degradation with age as proposed previously in mammals (*de Magalhães et al., 2009*). Moreover, expression of 29 lysosome-related genes is unaltered in aging adults (*Figure 5A* and *Supplementary file 4*). Thus, the overall profile of lysosomal transcripts is obviously remodeled, but not all lysosomal gene expression patterns are altered during aging.

## Lysosomes are modulated by multiple longevity pathways

We found that long-lived mutants representing three different longevity pathways all exhibited increased activity and better maintenance of lysosomes with age. Reducing IIS by the *daf-2* mutation suppresses age-associated lysosomal changes. *daf-2* lysosomes maintain their vesicular morphology, ultrastructure, high motility, acidity and degradation activity with age. The maintenance of lysosome activity with age is achieved at least in part through transcriptional regulation of lysosome genes. Loss of *daf-16* and *skn-1* reduces lysosome gene expression in *daf-2* and causes decreased lysosomal acidity and degradation activity. In addition to modulating lysosome gene expression, reducing IIS increases stress resistance and reduces cellular damage (*Shore and Ruvkun, 2013*). This may reduce substrate loading into lysosomes and thus help to maintain lysosome activity with age. Consistent with this, we found previously that loss of *daf-2* increases stress resistance in the lysosome-defective mutant *scav-3* and suppresses the membrane integrity defects in *scav-3* (*Li et al., 2016*). In addition to the IIS pathway, lysosomes are also modulated by caloric restriction and mitochondrial pathways. In the feeding-defective mutant *eat-2* and the mitochondrial mutant *isp-1*, appearance of age-related lysosomal changes is suppressed or delayed, and lysosome gene expression is increased. Thus, lysosomes may serve as a common target of multiple longevity pathways. Notably, only 2 out of the 43 lysosomal genes that are downregulated with age are targeted by all three pathways (*Figure 6—figure supplement 1A*). The IIS and caloric restriction pathways seem to target different sets of lysosome genes, whereas genes upregulated in *isp-1* mutants are mostly shared with the IIS pathway (*Figure 6—figure supplement 1A*). Future studies are needed to understand why and how lysosomal genes are selectively regulated by different pathways.

We identified DAF-16 and SKN-1 as key factors involved in modulating lysosome gene expression by multiple longevity pathways. By contrast, PHA-4 and HIF-1, the key downstream effectors of the caloric restriction and mitochondrial pathways, respectively, are dispensable for lysosome regulation. DAF-16 is reported to regulate lysosomal pH in the intestine in response to the reproductive cycle (*Baxi et al., 2017*). In this process, DAF-16 is activated by the DAF-9/Cytochrome P450 and DAF-

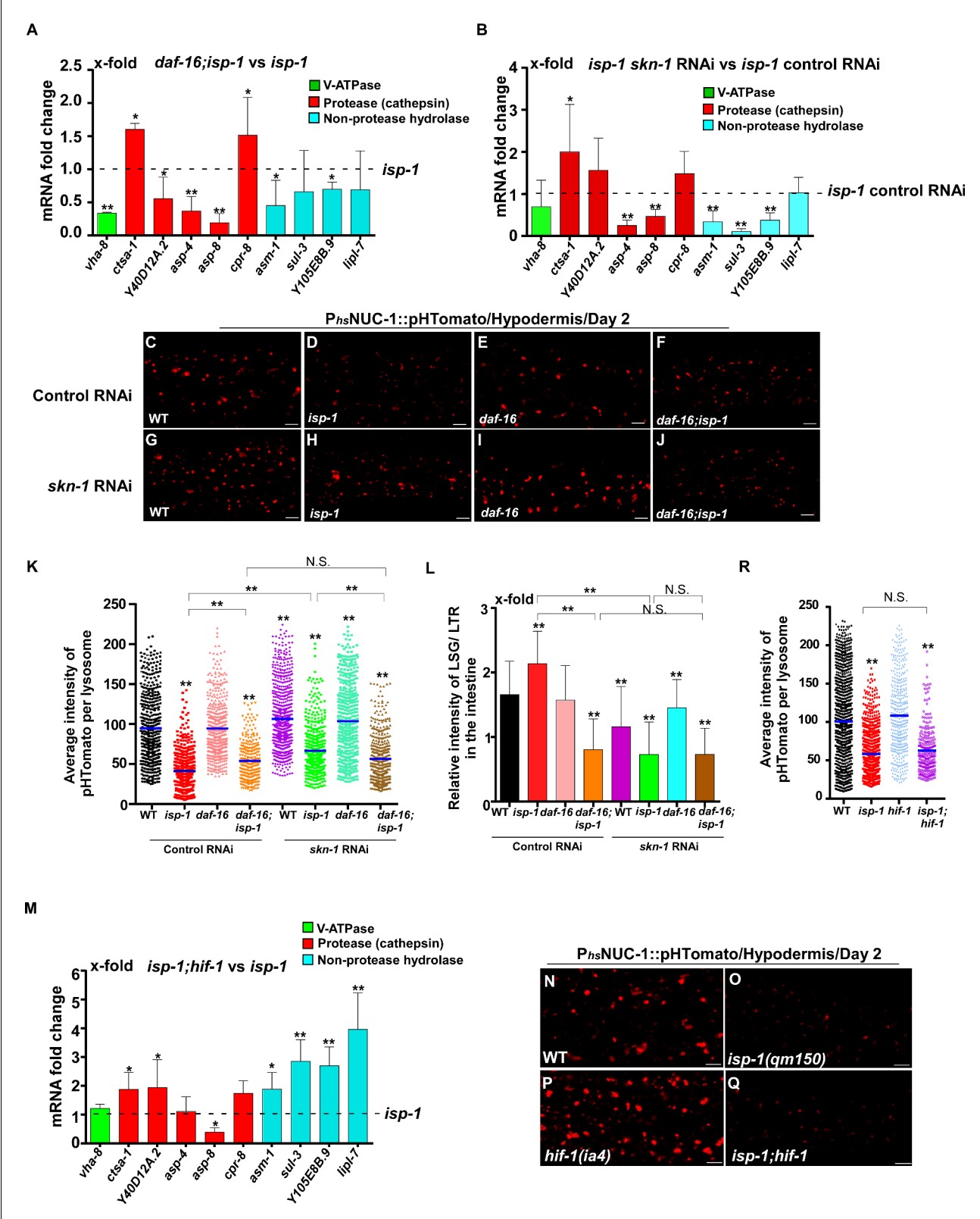

**Figure 8.** DAF-16 and SKN-1, but not HIF-1, regulate lysosomal acidity and gene expression in *isp-1* mutants. (**A, B, M**) Expression of the 10 upregulated lysosomal genes in *isp-1(qm150)* was analyzed by qRT-PCR in *daf-16;isp-1* (**A**), *isp-1 skn-1* RNAi (**B**) and *isp-1;hif-1* (**M**) worms at day 1. Three independent experiments were performed. The transcription level of lysosomal genes in *isp-1(qm150)* or *isp-1(qm150)* control RNAi at day 1 was normalized to '1' for comparison. (**C–J, N–Q**) Confocal fluorescence images of the hypodermis in the indicated strains expressing NUC-1::pHTomato
*Figure 8 continued on next page*

*Figure 8 continued*

controlled by the heat-shock (hs) promoter. Scale bars: 5 µm. The average intensity of pHTomato per lysosome was quantified (**K, R**). At least 20 animals were scored in each strain. (**L**) The relative intensity of LSG/LTR in the intestine was quantified in the indicated strains at day 2. At least 10 animals were scored in each strain. In (**A, B, K, L, M, R**), data are shown as mean ± SD. Multiple *t* testing (**A, B, M**) or one-way ANOVA with Tukey's multiple comparisons test (**K, L, R**) was performed to compare datasets of double mutants with *isp-1* (**A, M**), or *isp-1* control RNAi (**B**), or to compare all other datasets with wild type (**R**) or with wild type treated with control RNAi (**K, L**), or to compare datasets that are linked by lines (**K, L, R**). *p<0.05; **p<0.001. All other points had p>0.05. N.S., no significance.

The online version of this article includes the following source data for figure 8:

**Source data 1.** Numerical data that are represented as a bar graph in *Figure 8A,B,K–M,R*.

12/Vitamin D receptor steroid signaling pathway in the gonad, which leads to increased expression of V-ATPase genes (*Baxi et al., 2017*). In addition, microarray analyses identified several *vha* genes that are upregulated by SKN-1 under non-stress conditions (*Oliveira et al., 2009*). Here we found that loss of *daf-16* and *skn-1* reduces expression of lysosome genes that encode membrane proteins, hydrolases and V-ATPase subunits in *daf-2* and *isp-1*, and these two mutations affect lysosomal acidity and/or degradation activity in a non-additive manner. This suggests that DAF-16 and SKN-1 act in concert to modulate lysosome activity in response to reduced IIS and impaired mitochondrial function. On the other hand, DAF-16 and SKN-1 appear to act in parallel to maintain expression of *vha-12* and *vha-5* in *eat-2* mutants, and loss of their function affects the acidity of *eat-2* lysosomes in an additive manner. Further studies are needed to understand how DAF-16 and SKN-1 cooperate to modulate lysosome gene expression in different conditions.

The TFEB ortholog HLH-30 influences lifespan extension by multiple pathways via its role in autophagy and lipophagy, but its functions are highly context-dependent (*O'Rourke and Ruvkun, 2013*, *Lapierre et al., 2013*; *Dall and Færgeman, 2019*). It was reported recently that DAF-16 and HLH-30 act as a complex to co-regulate longevity-promoting genes in IIS mutants (*Lin et al., 2018*). Consistent with this, we found that expression of 6 lysosomal hydrolase genes in *daf-2* is reduced by loss of *hlh-30* and 5 of them are also targeted by DAF-16. The other 8 DAF-16-regulated lysosome genes are not affected by *hlh-30* mutation. The lysosome degradation activity in *daf-2* worms, however, seems to be unaffected by *hlh-30* mutation, and is higher in *daf-16;daf-2;hlh-30* than in *daf-16;daf-2*. We suspect that loss of *hlh-30* causes a decrease in the autophagy level, which may have a beneficial effect on lysosomal activity due to reduced cargo loading into lysosomes.

## Lysosome function is essential for lifespan extension

Our data indicate that lysosome function is essential for lifespan extension induced by multiple mechanisms. Maintenance of lysosome activity and dynamics may promote degradation of lipids, misfolded proteins and damaged organelles, which all accumulate with age. Notably, autophagy capacity declines with age in several species, which may be attributed to impaired activation and progression of autophagy and/or a decline in degradation of autophagic cargo in lysosomes (*Hansen et al., 2018*). On the other hand, autophagy activity increases in multiple long-lived mutants and is important for lifespan extension (*Meléndez et al., 2003*; *Hansen et al., 2008*; *Lapierre et al., 2013*; *Tóth et al., 2008*). It is conceivable that longevity pathways upregulate the functionality of both autophagy and lysosomes to achieve efficient cellular clearance for lifespan extension. However, autophagy and lysosomes may be differentially regulated by longevity pathways. For example, DAF-16 is not required for the increased level of autophagy in *daf-2* (*Hansen et al., 2008*), but is important for lysosome regulation. Moreover, PHA-4 is required for the elevated autophagy in *eat-2* mutants (*Hansen et al., 2008*), but is dispensable for the upregulation of lysosomal activity. mTORC1 inhibits autophagy activity but is important for lysosomal tubulation in the reformation process and for LPS-induced tubulation of lysosomes in macrophages and DCs (*Yu et al., 2010*; *Saric et al., 2016*; *Hipolito et al., 2019*). Inhibition of TORC1 has no effect on either appearance or enrichment of tubular lysosomes in aged *C. elegans* (our unpublished results). Thus, TORC1 activity may not be required for age-associated lysosomal tubule formation in worms. Future investigations are needed to understand how lysosomes are reshaped during aging and how the regulation of lysosomes and autophagy is coordinated in different longevity-promoting pathways. It is worth noting that in our study, the age-associated alterations in lysosomal morphology, motility and acidity were mainly examined in hypodermal and intestinal cells, which are big and amenable to cell biology

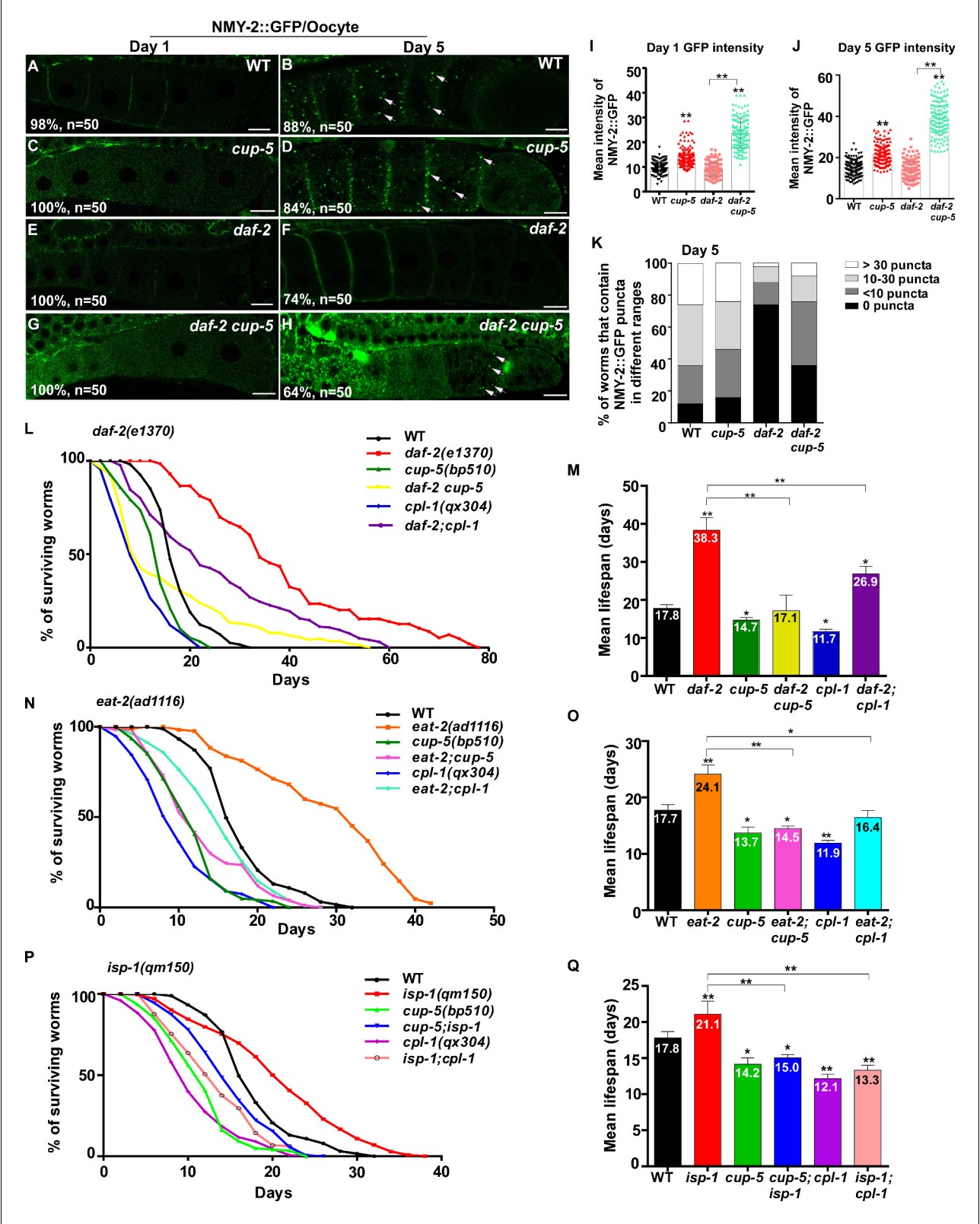

**Figure 9.** Lysosome activity is important for clearance of aggregate-prone proteins and lifespan extension. (A–H) Confocal fluorescence images of the oocytes in wild type (WT; A, B), *cup-5(bp510)* (C, D), *daf-2(e1370)* (E, F) and *daf-2 cup-5* (G, H) expressing NMY-2::GFP at different ages. White arrows indicate NMY-2::GFP puncta. Scale bars: 10 μm. (I–K) The average intensity of NMY-2::GFP (I, J) and the number of NMY-2::GFP puncta (K) were quantified. 50 animals were scored in each strain. (L–Q) Lifespan analyses were performed in the indicated strains. More than 100 worms were

*Figure 9 continued on next page*

*Figure 9 continued*

examined in each strain and three independent experiments were performed. The mean lifespan in the indicated strains was quantified and is shown in (M, O, Q). In (I, J, M, O, Q), data are shown as mean ± SD. One-way ANOVA with Tukey's multiple comparisons test (I, J) or multiple *t* testing (M, O, Q) was performed to compare all other datasets with wild type, or datasets that are linked by lines. *p<0.05; **p<0.001. All other points had p>0.05. The online version of this article includes the following source data and figure supplement(s) for figure 9:

**Source data 1.** Numerical data that are represented as a bar graph in *Figure 9I–Q*.

**Figure supplement 1.** Lysosome activity is important for clearance of aggregate-prone proteins in *eat-2* and *isp-1* mutants.

**Figure supplement 1—source data 1.** Numerical data that are represented as a bar graph in *Figure 9—figure supplement 1M–O,X and Z*.

analysis. We have not been able to examine the age-related changes in lysosomal properties in small-sized cells such as neurons. Further studies are required to understand whether lysosomes make tissue-specific contributions to aging and lifespan extension.

# Materials and methods

## Key resources table

| Reagent type (species) or resource | Designation | Source or reference | Identifiers | Additional information |
|---|---|---|---|---|
| Strain (*C. elegans*) | N2 | CGC | RRID:WB-STRAIN:N2_(ancestral) | wild type (Bristol) |
| Strain (*C. elegans*) | CF1038 | DOI: 10.1126/science.1083701 | RRID:WB-STRAIN: WBStrain00004840 | *daf-16(mu86)* |
| Strain (*C. elegans*) | PS3553 | DOI: 10.1126/science.1083701 | RRID:WB-STRAIN: WBStrain00030901 | *hsf-1(sy441)* |
| Strain (*C. elegans*) | DA1116 | DOI: 10.1073/pnas.95.22.13091 | RRID:WB-STRAIN: WBStrain00005548 | *eat-2(ad1116)* |
| Strain (*C. elegans*) | CF1041 | DOI: 10.1126/science.1139952 | RRID:WB-STRAIN: WBStrain00006375 | *daf-2(e1370ts)* |
| Strain (*C. elegans*) | HZ108 | DOI: 10.4161/auto.7.11.17759 | | *cup-5(bp510)* |
| Strain (*C. elegans*) | QV225 | DOI: 10.1534/g3.115.023010 | RRID:WB-STRAIN: WBStrain00031273 | *skn-1(zj15)* |
| Strain (*C. elegans*) | MQD887 | DOI: 10.1016/s1534-5807 (01)00071–5 | RRID:WB-STRAIN: WBStrain00026670 | *isp-1(qm150)* |
| Strain (*C. elegans*) | FX01978 | Shohei Mitani | RRID:WB-STRAIN: WBStrain00022468 | *hlh-30(tm1978)* |
| Strain (*C. elegans*) | XW10101 | DOI: 10.1091/mbc.E14-01-0015 | | *cpl-1(qx304)* |
| Strain (*C. elegans*) | ZG31 | DOI: 10.1016/j.cub.2010.10.057 | RRID:WB-STRAIN: WBStrain00040824 | *hif-1(ia4)* |
| Strain (*C. elegans*) | XW5399 | DOI: 10.1126/science.1220281 | | *qxIs257* ($P_{ced-1}$NUC-1::CHERRY) |
| Strain (*C. elegans*) | XW8056 | DOI: 10.1083/jcb.201602090 | | *qxIs430* ($P_{scav-3}$SCAV-3::GFP) |
| Strain (*C. elegans*) | XW10197 | DOI: 10.1016/j.devcel.2019.10.020 | | *qxIs468* ($P_{myo-3}$LAAT-1::GFP) |
| Strain (*C. elegans*) | XW11282 | DOI: 10.1016/j.devcel.2019.10.020 | | *qxIs520* ($P_{vha-6}$LAAT-1::GFP) |
| Strain (*C. elegans*) | XW13734 | DOI: 10.1016/j.devcel.2019.10.020 | | *qxIs612* ($P_{hs}$NUC-1::sfGFP ::CHERRY) |
| Strain (*C. elegans*) | XW19180 | this paper | | *qxIs750* ($P_{hs}$NUC-1::pHTomato) |

*Continued on next page*

*Continued*

| Reagent type (species) or resource | Designation | Source or reference | Identifiers | Additional information |
|---|---|---|---|---|
| Strain (*C. elegans*) | JJ1473 | DOI: 10.1242/dev.00735 | RRID:WB-STRAIN: WBStrain00022491 | *zuIs45* ($P_{nmy-2}$NMY-2::GFP) |
| Bacterial and virus strains | Vidal RNAi library | Open Biosystems | ORF RNAi collection V2 | *pha-4* and *skn-1* |
| Antibody | anti-CPL-1 (rat polyclonal) | DOI: 10.1126/science.1220281 | | WB(1:1000) |
| Antibody | anti-alpha-Tubulin (mouse monoclonal) | Sigma-Aldrich (Missouri, USA) | Cat #T5168; RRID:AB_477579 | WB(1:10000) |
| Antibody | anti-CHERRY (mouse monoclonal) | SUNGENE BIOTECH (Tianjin,China) | Cat#KM8017 | WB(1:1000) |
| Recombinant DNA reagent | pPD49.26-$P_{hs}$ NUC-1::pHTomato | this paper | | Cloning described in 'Plasmid construction' |
| Sequence-based reagent | pHTomato S KpnI_MluI | This paper | PDFZ1322 | cgcgGGTACCg gaACGCGTATG ATCAAGGAGT TCATGCGCTTC |
| Sequence-based reagent | pHTomato CAS SacI_NotI | This paper | PDFZ1323 | cgcgGAGCTC GCGGCCGC TTACTGTGCC TCCGCTGGCGC |
| Sequence-based reagent | Other primers used in this paper, see **Supplementary file 6** | This paper | | |
| Chemical compound, drug | LysoTracker Red DND-99 | Invitrogen (Oregon, USA) | Cat #L7528 | |
| Chemical compound, drug | LysoSensor Green DND-189 | Invitrogen (Oregon, USA) | Cat #L7535 | |
| Chemical compound, drug | Trizol | Invitrogen (Oregon, USA) | 15596–018 | |
| Commercial assay or kit | PrimeScript RT Reagent Kit | TaKaRa | RR037A | |
| Commercial assay or kit | FS Universal SYBR Green Master | Roche | 4913850001 | |
| Commercial assay or kit | SuperSignal West Pico PLUS. Chemiluminescent Substrate | ThermoFisher | 34577 | |
| Software, algorithm | Volocity | PerkinElmer (Massachusetts, USA) | RRID:SCR_002668 | |
| Software, algorithm | Zen | Carl Zeiss (Oberkochen, Germany) | RRID:SCR_01367 | |
| Software, algorithm | Image J | N/A | V1.42q, RRID:SCR_003070 | |

## *C. elegans* strains

Strains of *C. elegans* were cultured and maintained using standard protocols (*Brenner, 1974*) unless indicated otherwise. The N2 Bristol strain was used as the wild type (WT) strain Genome-integrated arrays (*qxIs*) were acquired by γ-irradiation to achieve stable expression from arrays with low copy numbers. The following strains were used in this work: linkage group (LG) I, *daf-16(mu86), hsf-1 (sy441)*; LG II, *eat-2(ad1116)*; LG III, *daf-2(e1370ts), cup-5(bp510)*; LG IV, *skn-1(zj15), isp-1(qm150), hlh-30(tm1978)*; LG V, *cpl-1(qx304), hif-1(ia4)*. The reporter strains used in this study include *qxIs257* ($P_{ced-1}$NUC-1::CHERRY), *qxIs468* ($P_{myo-3}$LAAT-1::GFP), *qxIs520* ($P_{vha-6}$LAAT-1::GFP), *qxIs750* ($P_{hs}$NUC-1::pHTomato), *qxIs612* ($P_{hs}$NUC-1::sfGFP::CHERRY), *zuIs45* ($P_{nmy-2}$NMY-2::GFP).

## Microscopy and imaging analysis

Differential interference contrast (DIC) and fluorescence images were captured with an Axioimager A1 (Carl Zeiss) equipped with epi-fluorescence [Filter Set 13 for GFP (excitation BP 470/20, beam splitter FT 495, emission BP 503–530) and Filter Set 20 for Cherry (excitation BP 546/12, beam splitter FT 560, emission BP 575–640)] and an AxioCam monochrome digital camera (Carl Zeiss). Images were processed and viewed using Axio-vision Rel. 4.7 software (Carl Zeiss). A 63 × objective (Plan-Neofluar NA1.30) was used with Immersol 518F oil (Carl Zeiss). Confocal images were captured by a Zeiss 880 inverted laser scanning confocal microscope with 488 nm (emission filter BP 503–530) and 543 nm (emission filter BP 560–615) lasers, and images were processed and viewed using Zen software (Carl Zeiss). All images were taken at 20˚C.

## Time-lapse recording using spinning-disk microscopy

C.C. elegans adults at different ages (days 1, 3, 5, 9) were mounted on agar pads in M9 buffer with 5 mM levamisole to prevent movement of the animals. Fluorescence images were captured using a 60 × objective (CFI Plan Apochromat Lambda; NA 1.45; Nikon) with immersion oil (type NF) on an inverted fluorescence microscope (Eclipse Ti-E; Nikon) with a spinning disk confocal scanner unit (UltraView; PerkinElmer) with 488 nm [emission filter 525 (W50)] and 561 nm [dual-band emission filter 445 (W60) and 615 (W70)] lasers. To follow lysosomal dynamics in worms expressing NUC-1::CHERRY, images were captured every 1 s for 1–2 min. The collected images were viewed and analyzed using Volocity software (PerkinElmer).

## RNAi treatment

RNAi was performed by using the standard feeding method and Vidal RNAi library (Open biosystem) (*Rual et al., 2004*). For most experiments, 3–5 L4 larvae (P0) were cultured on the RNAi plate and F1 progeny at late larval and young adult stages were examined. The *pha-4* and *skn-1* RNAi led to death of the F1 progeny. In this case, ~50 bleached L1 larvae were transferred to plates seeded with bacteria expressing either control double stranded RNA (dsRNA; L4440 empty vector; Control RNAi) or dsRNA corresponding to *pha-4* and *skn-1*. The phenotype was examined at adult stages in the same generation.

## Quantification of lysosomal tubule length

Fluorescence images of *C. elegans* adults at different ages (days 1, 3, 5, 9) expressing NUC-1::CHERRY were captured by laser scanning confocal microscopy (Carl Zeiss). The length of NUC-1::CHERRY-positive tubules in each worm was quantified by Image J software. Tubular lysosomes that crossed one another were counted as two individual tubules. 10 lysosomal tubules were measured in each animal and at least 20 animals were scored in each strain at each day.

## Quantification of lysosome number and volume

Fluorescence images of *C. elegans* adults at different ages (days 1, 3, 5, 9) expressing NUC-1::CHERRY in 10–15 z-series (0.5 µm/section) were captured by spinning-disk microscopy. Serial optical sections were analyzed, and the volume and number of NUC-1::CHERRY-positive vesicular lysosomes per unit area (31 × 43 µm²) was quantified by Volocity software (PerkinElmer). At least eight animals were quantified in each strain at each stage. The total volume of vesicular and tubular lysosomes was quantified by Volocity. At least 10 worms were quantified in each strain at each day.

## Quantification of lysosome dynamics

Time-lapse images of *C. elegans* L4-stage larvae and adults at different ages (days 1, 3, 5, 9) expressing NUC-1::CHERRY were captured by spinning-disk microscopy. To quantify Pearson's correlation coefficient, the colocalization of two frames taken 60 s apart was analyzed by Volocity software (PerkinElmer). The average velocity (displacement rate) of tubular and vesicular lysosomes within 60 s was measured by Volocity software (PerkinElmer). At least 10 independent videos were recorded and quantified in each strain at each day.

## LysoSensor green and LysoTracker staining

C.C. elegans adults at different age (~40 at each age) were soaked in 80 µl M9 buffer containing LysoSensor Green DND 189 and LysoTracker Red DND 99 at 10 µM for staining in the intestine and 60 µM for staining in the hypodermis (Invitrogen, Oregon, USA). Staining was carried out for 1 hr at 20°C in the dark. Worms were then transferred to NGM plates with fresh OP50 and allowed to recover at 20°C for 1 hr in the dark before examination. The relative intensity of LSG/LTR was quantified by Volocity (PerkinElmer).

## Quantification of NUC-1::pHTomato intensity

C. elegans adults (1 day post L4/adult molt) expressing $P_{hs}$NUC-1::pHTomato were incubated at 33°C for 30 min and recovered at 20 °C for 24 hr before examination. The average intensity of pHTomato per lysosome in the hypodermis was measured by Volocity (PerkinElmer). At least 20 worms were quantified in each strain.

## Lysosome degradation activity assay
### Examination and quantification of CPL-1 processing

About 50 C. elegans adults at different ages (days 1, 5, 9) were picked and washed three times with M9. The worms were lysed by boiling followed by several rounds of freezing and thawing. The resulting worm lysate was resolved by SDS-PAGE and the CPL-1 processing was detected by anti-CPL antibodies (Antibody core, NIBS, 1:1000). $\alpha$-tubulin antibody (Sigma) was used at 1:5000 as an internal control. The band intensities of the mature and pro- forms of CPL-1 were quantified by Image J software, then CPL-1 processing was quantified by dividing the mature CPL-1 by the total CPL-1 (both pro- and mature forms). three independent experiments were performed and quantified in each strain at each stage.

## Quantification of NUC-1::CHERRY cleavage

Adult worms (~50, 1 day post L4/adult molt) expressing NUC-1::CHERRY were washed three times in M9. The worms were lysed by boiling followed by several rounds of freezing and thawing. The resulting worm lysate was analyzed by Western blot using anti-CHERRY antibodies (SUNGENE BIOTECH, China, 1:1000) and anti-tubulin antibodies (Sigma, 1:5000). The intensities of NUC-1::CHERRY and CHERRY bands were quantified by Image J software and the extent of cleavage was calculated by dividing the amount of CHERRY by the total amount of NUC-1::CHERRY and CHERRY. three independent experiments were performed and quantified in each strain.

## HVEM analysis

C. elegans adults at different ages (days 1, 5) were rapidly frozen using a high-pressure freezer (EM PACT2; Leica Biosystems). Freeze substitution was performed in anhydrous acetone containing 1% osmium tetroxide. The samples were kept sequentially at −90°C for 72 hr, −60°C for 8 hr, and −30°C for 8 hr and were finally brought to 20°C for 10 hr in a freeze-substitution unit (EM AFS2; Leica Biosystems). The samples were washed three times (1 hr each time) in fresh anhydrous acetone and were gradually infiltrated with Embed-812 resin in the following steps: resin/acetone 1:3 for 3 hr, 1:1 for 5 hr, 3:1 overnight, and 100% resin for 4 hr. Samples were then kept overnight and embedded at 60°C for 48 hr. The fixed samples were cut into 70 nm sections with a microtome EM UC7 (Leica Biosystems) and electron-stained with uranyl acetate and lead citrate. Sections were observed with a JEM-1400 (JEOL) operating at 80 kV. For quantitative analysis of lysosomes, three to five animals were analyzed in each strain at each stage, using eight 70 nm sections (non-consecutive sections, spaced at 5000 nm) in each animal. Images of each lysosome were taken at high magnification (60,000 × or 30,000×) and the numbers were counted manually. Lysosome diameter was measured by Image J software.

## Quantitative real-time PCR (qRT-PCR)

Worms were synchronized and cultured at 20°C to different ages (adult day 1 and day 5). Total RNA was extracted from 20 µl worm pallets at each stage using Trizol (Invitrogen/Life Technologies, Carlsbad, CA) and reverse transcribed by a PrimeScript RT Reagent Kit (TaKaRa). The reverse transcription products (cDNA) were diluted to 10 ng/µl and used as the template for quantitative PCR. For

quantitative RT-PCR, custom-designed primers were mixed with SYBR Green Mix (Roche) and samples were analyzed using a PCR biosystems QuantStudio 7 Flex (Applied Biosystems). The gene *cdc-42* was used as the internal reference. At least three independent experiments were performed with three replications each time.

## Quantification of NMY-2::GFP intensity and number of puncta

Fluorescence images of *C. elegans* adults expressing NMY-2::GFP at different ages (days 1 and 5) were captured by laser scanning confocal microscopy (LSM 880, Carl Zeiss). Fluorescence intensity in oocytes (the second, third and fourth oocytes counted from the spermatheca) were measured by Volocity software. The number of NMY-2::GFP puncta in oocytes was counted manually. 50 animals were quantified in each strain at each day.

## Lifespan assay

Worms were synchronized and cultured at 20°C until they reached the L4 stage. About 150 L4-stage worms (day 0) were picked to NGM plates with fresh OP50, 15 worms per plate. Worms were considered dead when they failed to respond to gentle touches on the head and tail with a worm picker. The surviving worms were counted every 2 days and were transferred to new plates to avoid interference from the progeny. Animals that crawled off the plate, exploded, bagged, or became contaminated were discarded. At least 100 worms were quantified in each strain. At least three independent experiments were performed for each strain. Representative survival curves are shown in *Figure 9L,N,P* and the mean lifespan from three experiments is shown in *Figure 9M,O,Q*.

## Plasmid construction

To generate $P_{hs}$NUC-1::pHTomato, pHTomato was amplified from plasmid $P_{mito}$pHTomato (Chen Chang Lab, Institute of Biophysics, Chinese Academy of Science, China) using primers PDFZ1322/PDFZ1323 and was ligated to pPD49.26-$P_{hyp-7}$NUC-1 through the Kpn I-Mlu I/Sac I sites, followed by replacement of the *hyp-7* promoter with the heat-shock promoter (hs) through the BamH I site.

## Statistical analysis

The standard deviation (SD) was used as y-axis error bars for bar charts plotted from the mean value of the data. Data derived from different genetic backgrounds and/or different stages were compared by Multiple *t* testing, paired *t* testing, one-way ANOVA with Tukey's multiple comparisons test or two-way ANOVA with Fisher's LSD test. Data were considered statistically different when $p<0.05$. $p<0.05$ is indicated with single asterisks, $p<0.001$ with double asterisks.

## Acknowledgements

We thank Dr. Mengqiu Dong for discussion and critical reading of the manuscript and Dr. Isabel Hanson for editing services. Some strains were provided by the CGC, which is funded by NIH Office of Research Infrastructure Programs (P40OD010440). This work was supported by the Ministry of Science and Technology (2016YFA0500203), the National Natural Science Foundation of China (3163001, 91754203) and the Strategic Priority Research Program of the Chinese Academy of Sciences (XDB19000000) to X.W.The authors declare no competing financial interests.

## Additional information

### Competing interests

Xiaochen Wang: Reviewing editor, *eLife*. The other authors declare that no competing interests exist.

## Funding

| Funder | Grant reference number | Author |
|---|---|---|
| Ministry of Science and Technology of the People's Republic of China | 2016YFA0500203 | Xiaochen Wang |
| National Natural Science Foundation of China | 3163001 | Xiaochen Wang |
| National Natural Science Foundation of China | 91754203 | Xiaochen Wang |
| Chinese Academy of Sciences | Strategic Priority Research Program XDB19000000 | Xiaochen Wang |

The funders had no role in study design, data collection and interpretation, or the decision to submit the work for publication.

## Author contributions
Yanan Sun, Investigation, Visualization, Writing - original draft; Meijiao Li, Investigation, Visualization; Dongfeng Zhao, Investigation; Xin Li, Investigation, Writing - original draft; Chonglin Yang, Supervision, Writing - review and editing; Xiaochen Wang, Conceptualization, Supervision, Funding acquisition, Writing - original draft

## Author ORCIDs
Xiaochen Wang (iD) https://orcid.org/0000-0002-4344-0925

## Decision letter and Author response
Decision letter https://doi.org/10.7554/eLife.55745.sa1
Author response https://doi.org/10.7554/eLife.55745.sa2

# Additional files

## Supplementary files
- Supplementary file 1. The 85 lysosome-related genes analyzed by RT-PCR.
- Supplementary file 2. Expression of 43 lysosomal genes is reduced in wild type (WT) at day 5.
- Supplementary file 3. Expression of 13 lysosomal genes is increased in wild type (WT) at day 5.
- Supplementary file 4. Expression of 29 lysosome genes is unaltered in wild type (WT) at day 5.
- Supplementary file 5. Lysosome gene expression is upregulated in *daf-2*, *eat-2* and *isp-1* mutants.
- Supplementary file 6. Primers used for quantitative RT-PCR, related to key resources table.
- Transparent reporting form

## Data availability
All data generated or analyzed during this study are included in the manuscript and supporting files.

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
