## [Decision Letter]

**Acceptance summary:**

Lysosome dysfunction has been proposed to occur during aging and there is good evidence for the same in yeast. Taking advantage of the *C. elegans* mutants that have increased lifespan, this paper provides convincing evidence that long-living mutants retain lysosome function during aging, unlike their wild-type counterparts. Indeed, increased lifespan of these mutants was reduced upon disruption of lysosome function, suggesting that maintenance of lysosomal activity is required for lifespan extension. The authors demonstrate that expression of lysosomal genes including, cathepsins and V-ATPases is regulated by transcription factors that act downstream of multiple longevity pathways. This study lays the groundwork towards future research in understanding how organelle composition and function changes during aging and the impact of these on organismal lifespan.

**Decision letter after peer review:**

Thank you for submitting your article "Lysosome activity is modulated by multiple longevity pathways and is important for lifespan extension in *C. elegans*" for consideration by *eLife*. Your article has been reviewed by three peer reviewers, one of whom is a member of our Board of Reviewing Editors, and the evaluation has been overseen by David Ron as the Senior Editor. The reviewers have opted to remain anonymous.

The reviewers have discussed the reviews with one another and the Reviewing Editor has drafted this decision to help you prepare a revised submission.

Summary:

In their manuscript, Sun et al. present findings that aging in *C. elegans* leads to reduced acidity, degradative capacity, and motility of lysosomes, as well as transition from vesicular to tubular lysosomes. Interestingly, long-living *C. elegans* mutants (*daf-2*, *eat-2* and *isp-1*) resist this abatement in lysosome function, providing a correlation between longevity and lysosome activity. The authors find that expression of lysosome-related genes (prominent among these are V-ATPases and cathepsins) is reduced with aging. Transcription factors DAF-16/FOXO and SKN-1/NRF2 were required for maintaining lysosomal gene expression and lysosome function in the long-lived mutants. Finally, the authors show that long-living mutants have a higher propensity to clear protein aggregates and that this depended on DAF-16 or SKN-1. Consistent with the idea that lysosome function declines with age, disruption of lysosome function reduced the lifespan of *C. elegans* and "cancelled" the long lifespan of long-living mutants such as *daf-2*.

Overall, this is a very interesting study, well written and timely that reveals age-associated decline in lysosomal function and the protective effects of longevity mechanisms against this decline.

The reviewers raise few concerns that must be adequately addressed before the paper can be accepted for publication in *eLife*. The reviewers have also suggested changes in manuscript text to avoid extrapolation of the findings presented in the manuscript. Some of the revisions require additional experimentation within the scope of the presented studies and techniques.

Essential revisions:

1) Referring to lysosome tubulation as a process of lysosome reformation is problematic, as lysosome tubulation may be a reformation process (as described in Yu et al.) or could also be part of other types of lysosome dynamics that mediate cargo exchange, sorting of cargo, and or other types of function. For example, mammalian macrophages and dendritic cells exposed to LPS become highly tubular (Vyas et al., 2007; and Saric et al., 2016), while tubules form between phagosomes to help mediate cargo exchange (Mantegazza et al., 2014). Additionally, Hipolito et al., 2020, shows that lysosomes undergo remodelling both in shape, size, and the transcription profile encoding lysosomal transcripts in response to macrophage activation. This is not likely an issue of lysosome reformation. Thus, the authors should alter the text to be more nuanced in relation to lysosome tubulation. Please avert referring to tubular lysosomes as "reforming" or "pre-reforming" – this is strictly an assumption at this stage and the authors provide no evidence to support this.

2) The authors mention in the Discussion "the lysosomal tubules in aged adults are static and are not readily stained by LysoSensor Green" indicating that "the lysosomal tubules enriched in aged adults are probably catalytically inactive". This should be investigated and quantification of LysoSensor Green staining in tubular lysosomes should be presented.

3) Figure 4A-L should include quantification of tubular lysosome length. It is also unclear how the tubules for quantification were chosen (Materials and methods) and more details should be provided in the Materials and methods section.

4) The conclusions on the degradation capacity of lysosomes are mostly based on processing of protein substrate Nuc1-Cherry and CPL-1 processing. Thus, testing lysosomal degradative ability during aging by analysis of an endogenous cargo could be an important addition to the results. If probes such as Magic Red and/or DQ-BSA permeate through *C. elegans* surface, they can be also used to analyze lysosomal degradation capacity.

5) The authors also need to be more careful about referring to lysosome changes as en block. The manuscript text describes the changes to lysosomes caused by aging, long-living mutants, and those driven by *daf-16*/SKN1 as if all aspects of lysosome function and identity changed. This is clearly not the case as indicated by the RNA expression data presented in the manuscript. While a large number of transcripts encoding lysosomal proteins are reduced with age, there is a substantial number that increase or is unchanged. This is also true for all conditions they examined. In the revised version, the authors should include all lysosomal transcripts that have been analyzed in the supplementary files (and not only the transcripts whose levels are reduced). The findings presented here suggest that lysosomes are remodelled in some way but do not necessarily indicate total disruption of lysosomal function. The authors are thus asked to discuss with greater nuance this issue and to reflect this in the Abstract. This will avoid giving the impression that the lysosomes in their entirety are changing – certainly; this is not supported by their data.

6) The authors have employed mutants lacking CUP-5 and CPL-1 activity to conclude that lifespan extension in *C. elegans* requires lysosome function. However, whether decline in lysosomal function during aging is comparable to what is observed in the *cup-5* and *cpl-1* mutants is not clear. This can be addressed by functional assays such as cathepsin processing, LSG/LTR ratio tested in any one of the mutants (*cup-5*/*cpl-1*) and compared with worms at day 9.

7) The authors have shown that ratio of lysosensor to lysotracker is reduced in aging worms, indicating that lysosome become less acidic (Figure 2I). Further the pHTomato experiments also indicate that pH is less acidic in aged worms. It would be very informative if the authors can measure the actual pH of lysosomal compartments to understand the magnitude of pH change during aging. If the lysosomal pH measurement is not feasible in this model system, the findings with the pHTomato probe should be complemented with data that directly report pH change such the LSG/LTR ratio. The latter experiment can be added to Figure 6.

8) The discussion on metabolism and role in aging and lysosome function begs the question on what happens to mTORC1 activity. Do the long-living mutants have lower mTORC1 activity? If so, could this help explain lower tubulation in long-living *C. elegans*. This may be connected to mTOR modulation of lysosome tubulation and remodelling in mammalian macrophages exposed to LPS (Saric et al., 2016; Hipolito et al., 2020). If nothing else, this possibility could be discussed.

9) Figure 9C and G, the images of NMY-2::GFP do not appear to be representative for the quantification result shown Figure 9I. Further, NMY-2 GFP appears cytosolic in *daf-2cup-5* mutant (Figure 9H). It is not clear how the authors have quantified punctae (Figure 9K) in this mutant when so much of the protein appears cytosolic.

10) Total CPL-1 protein levels are increased upon aging in WT (Figures 2N and 4S) but, qPCR results in Figure 5C show reduced *cpl-1* expression at day 5. The authors should provide possible explanation for this difference (for instance, comment upon the CPL-1 protein stability).

11) From the data shown in Figures 5, 6 and 7, are there any common lysosomal genes that are upregulated in all the three longevity mutants while a corresponding decrease is observed in older worms. These should be highlighted and discussed in the Results.

12) The end of the Discussion is a bit abrupt. The authors can probably restate the importance of the findings and also it would be of service to the community to highlight specific caveats and shortcomings of the study to avoid excessive extrapolation in future interpretations.

13) For quantification and statistical analysis of the data, it is not clear to this reviewer if the authors examined all animals for a specific "read-out" in one day or during independent days. For example, the authors state "At least 10 worms were quantified in each strain at each stage" when describing the method to measure tubule length. Were 10 animals all done in one day or over different days? This comment applies elsewhere and it is important to know.

14) For quantitative real-time PCR (qRT-PCR) experiments, it is not clear whether RNA was pooled from multiple worms and if so please state the number.

15) For quantification of NMY-2::GFP intensity experiment, reasoning for using 20 worms for quantification at day 1 and 50 worms at day 5 is not clear.

---

## [Author Response]

Essential revisions:1) Referring to lysosome tubulation as a process of lysosome reformation is problematic, as lysosome tubulation may be a reformation process (as described in Yu et al.) or could also be part of other types of lysosome dynamics that mediate cargo exchange, sorting of cargo, and or other types of function. For example, mammalian macrophages and dendritic cells exposed to LPS become highly tubular (Vyas et al., 2007; and Saric et al., 2016), while tubules form between phagosomes to help mediate cargo exchange (Mantegazza et al., 2014). Additionally, Hipolito et al., 2020, shows that lysosomes undergo remodelling both in shape, size, and the transcription profile encoding lysosomal transcripts in response to macrophage activation. This is not likely an issue of lysosome reformation. Thus, the authors should alter the text to be more nuanced in relation to lysosome tubulation. Please avert referring to tubular lysosomes as "reforming" or "pre-reforming" – this is strictly an assumption at this stage and the authors provide no evidence to support this.

We agree with the reviewer that “correlation of tubular lysosomes in aged adults with lysosome reformation process” is strictly an assumption at this stage. In the revised manuscript, we have revised the parts of the text that describe tubule-containing lysosomes and we have discussed the point regarding formation of lysosomal tubules in aged adults.

2) The authors mention in the Discussion "the lysosomal tubules in aged adults are static and are not readily stained by LysoSensor Green" indicating that "the lysosomal tubules enriched in aged adults are probably catalytically inactive". This should be investigated and quantification of LysoSensor Green staining in tubular lysosomes should be presented.

As suggested by the reviewer, we performed LysoSensor Green (LSG) and LysoTracker Red (LTR) staining assays in the hypodermis at adult day 1 and day 5, when abundant vesicular and tubular lysosomes are observed, respectively. Consistent with staining in the intestine (Figure 2 A-D’’, I), we found that the fluorescence intensity ratio of LSG/LTR in hypodermal vesicular lysosomes is reduced significantly at day 5 (Figure 2—figure supplement 1A-A’’, C, revised manuscript). Moreover, the percentage of LTR-positive vesicular lysosomes stained by LSG is decreased at day 5 compared with day 1 (Figure 2—figure supplement 1A-A’’, C, revised manuscript). The lysosomal tubules in the hypodermis at day 5, however, were not stained by LSG and only weakly labeled by LTR (Figure 2—figure supplement 1B-B’’, D, revised manuscript). These data suggest that tubular lysosomal structures may be less acidic than the vesicular ones. These data are presented in Figure 2—figure supplement 1 in the revised manuscript.

3) Figure 4A-L should include quantification of tubular lysosome length. It is also unclear how the tubules for quantification were chosen (Materials and methods) and more details should be provided in the Materials and methods section.

In the original manuscript, the data for tubular lysosome length in *eat-2* and *isp-1* were presented in Supplementary Figure 3A. As suggested by the reviewer, these data are now presented in Figure 4M in the revised manuscript. To quantify tubular lysosome length, confocal fluorescence images of hypodermal lysosomes were taken, and individual lysosomal tubules were selected for quantification. If one lysosomal tubule crossed over another tubule, they were counted as two individual tubules. The length of each lysosomal tubule was quantified by Image J. In the original manuscript, 5 tubules were scored in each animal and 20 animals in total were quantified in each strain at each age. The results were presented in a scatter diagram (Supplementary Figure 3A in the original manuscript). In this diagram, each dot represents the average length of 5 lysosomal tubules in one animal and in total 20 dots (20 animals) were shown in each strain at each age. In the revised manuscript, we increased the sample size. We scored 10 individual lysosomal tubules in each animal and quantified 20 animals in each strain at each age. In the scatter diagram shown in Figures 1I and 4M in the revised manuscript, each dot represents the length of one individual tubule, and in total 200 dots (which represent the lengths of 200 tubules) are shown in each strain at each age. We have explained in detail how the lysosomal tubules are selected and quantified in the Materials and methods section.

4) The conclusions on the degradation capacity of lysosomes are mostly based on processing of protein substrate Nuc1-Cherry and CPL-1 processing. Thus, testing lysosomal degradative ability during aging by analysis of an endogenous cargo could be an important addition to the results. If probes such as Magic Red and/or DQ-BSA permeate through C. elegans surface, they can be also used to analyze lysosomal degradation capacity.

We understand the concern raised by the reviewers and agree that analysis of an endogenous lysosomal cargo is important for examining lysosomal degradation ability. In fact, we have tested several possibilities and found that NUC-1::CHERRY cleavage and CPL-1 processing are reliable assays for analyzing lysosomal degradation activity in *C. elegans*.

1) Magic Red and DQ-BSA are both protein substrates cleaved by cathepsins in lysosomes and are widely used to indicate lysosomal degradation activity in mammalian cells. They were also our first choice for testing lysosome degradation ability in worms. We have tried multiple ways to deliver the probes to lysosomes in live *C. elegans*. Unfortunately, none of the attempts worked. We cannot see Magic Red signal in worms by soaking or injection. DQ-BSA is not taken up efficiently by *C. elegans* even at high concentrations.

2) As we failed to use Magic Red or DQ-BSA as the lysosomal probe, we sought to find

*C. elegans* substrates whose cleavage in lysosomes can be followed directly and quantified to indicate lysosomal activity. Lysosomes degrade cargo delivered via endocytosis, phagocytosis or autophagy. As lysosomal degradation of phagocytic cargo, such as apoptotic cells, is difficult to follow and quantify directly, we focused on endocytic and autophagic cargoes. LGG-1 is a *C. elegans* homolog of Atg8/LC3, which associates with autophagic structures and intact autophagosomes that are delivered to lysosomes. We previously generated a single-copy insertion strain of GFP::LGG-1 by CRISPR-Cas9, in which GFP::LGG-1 is expressed at an endogenous level (Liu, et al., JCB 2018). By using this strain, we tested whether release of GFP from GFP::LGG-1, detected by Western blot, can be used to indicate lysosome degradation activity. As shown in Author response image 1, using whole worm lysates probed with anti-GFP antibodies, we observed processing of GFP::LGG-1 and release of free GFP. We found that GFP accumulates at a significantly higher level in the lysosome-defective mutants *cup-5* and *cpl-1* than in wild type (Author response image 1A, lanes 6 and 7 compared with lane 1, B). By contrast, no GFP accumulation was observed in *epg-6* mutants that block autophagosome formation (Author response image 1A, lane 5, B). These results suggest that GFP::LGG-1 is delivered to lysosomes through autophagy, followed by release and degradation of GFP in lysosomes. However, as the free GFP level is affected by both autophagy and lysosome activity, GFP release and/or degradation cannot be simply used to indicate lysosome degradation ability. This issue is particularly important when lysosome activity is analyzed during aging and in long-lived mutants. It is reported that autophagy activity declines with age but increases in long-lived *daf-2* worms (Chang et al., *eLife* 2017; Lapierre et al., Curr Biol., 2011; Melendez et al., 2003). We indeed observed reduced GFP accumulation in *daf-2* worms (Author response image 1A, lane 2, B), consistent with increased lysosome activity. However, it is unclear whether and how changes in autophagy contribute to the free GFP level in *daf-2*. In addition, GFP accumulation is reduced in *isp-1* but is unaltered in *eat-2* (Author response image 1A, lanes 3 and 4, B). Whether the changes in *isp-1* worms are caused by changes in autophagy, lysosome or both remains unclear.

3) Finally, we turned to cargoes delivered to lysosomes via the endocytic pathway. We considered that lysosomal hydrolases are probably the best candidates for this purpose because they are in fact endocytic cargoes destinated for lysosomes. To probe lysosomal activity, we utilized monomeric CHERRY fused to NUC-1, a lysosomal DNase II. The fluorescent protein CHERRY is not quenched in acidic conditions and is thus visible in lysosomes. Moreover, CHERRY can be cleaved from the fusion protein by lysosomal hydrolases, which requires the 11 N-terminal residues of CHERRY (Huang et al., PLOS one, 2014). Unlike GFP which is degraded fast in lysosomes, CHERRY is quite stable in lysosomes. Thus, the level of free CHERRY cleaved from NUC-1::CHERRY can be quantified to indicate lysosome degradation ability. In the previous study, we showed that CHERRY cleavage from NUC-1::CHERRY reduces in *cup-5*, which is defective in lysosomal degradation (Miao et al., 2020). Moreover, CHERRY cleavage increases significantly at molt when lysosomes are upregulated (Miao et al., 2020). In addition to NUC-1::CHERRY, which is expressed from multi-copy insertions, we tested whether processing of cathepsin L (CPL-1), which can be detected by anti-CPL-1 antibodies, may serve as a probe of lysosomal activity at an endogenous level. When delivered to lysosomes via endocytic transport, pro-CPL-1 is converted to the active mature form through proteolytic removal of the pro-domain in an autocatalytic manner or by other cathepsins (Stoka, Turk and Turk, 2016). Thus, processing of pro-CPL-1 to mature CPL-1 can be used to indicate lysosome degradation ability. In line with the NUC-1::CHERRY cleavage assay, CPL-1 processing is reduced significantly in *cup-5* mutants, consistent with defects in lysosome degradation activity (Figure 9—figure supplement 1). Moreover, *cpl-1* gene expression is not altered in the long-lived *daf-2*, *eat-2* and *isp-1* worms, which makes CPL-1 processing a suitable assay to examine lysosome activity in these mutants.

Currently, we are not aware of other endogenous cargoes whose delivery to and degradation in lysosomes have been clearly studied in worms. We will continue to optimize both NUC-1::CHERRY and CPL-1 processing assays and to develop other probes to examine lysosome degradation activity in *C. elegans*.

**Author response image 1. sa2fig1:** Examination of GFP::LGG-1 cleavage and degradation. (**A**) Western blot analysis of GFP::LGG-1 cleavage and degradation. Whole worm lysates were prepared in the indicated strains at adult day 3. (**B**) The free GFP level was quantified in the indicated strains. Three independent experiments were performed, and data are shown as mean ± SD. One-way ANOVA with Tukey’s multiple comparisons test was performed to compare all other datasets with wild type. **P*<0.05; ***P*<0.001. N.S., no significance.

5) The authors also need to be more careful about referring to lysosome changes as en block. The manuscript text describes the changes to lysosomes caused by aging, long-living mutants, and those driven by daf-16/SKN1 as if all aspects of lysosome function and identity changed. This is clearly not the case as indicated by the RNA expression data presented in the manuscript. While a large number of transcripts encoding lysosomal proteins are reduced with age, there is a substantial number that increase or is unchanged. This is also true for all conditions they examined. In the revised version, the authors should include all lysosomal transcripts that have been analyzed in the supplementary files (and not only the transcripts whose levels are reduced). The findings presented here suggest that lysosomes are remodelled in some way but do not necessarily indicate total disruption of lysosomal function. The authors are thus asked to discuss with greater nuance this issue and to reflect this in the Abstract. This will avoid giving the impression that the lysosomes in their entirety are changing – certainly; this is not supported by their data.

We apologize for not explaining our results more clearly in the original manuscript.

In the original manuscript, data for all 85 tested lysosome transcripts were included in the supplementary files. Supplementary file 1 shows the identity of all 85 lysosome-related genes that were tested; Supplementary file 2 shows qPCR data of the 43 lysosomal genes that are down regulated with age in wild type; Supplementary files 3 and 4 include qPCR data of the 13 and 29 lysosomal genes whose expression is increased and unaltered with age in wild type, respectively. Supplementary file 5 includes data of the 43 down-regulated lysosomal genes in the long-lived mutants *daf-2*, *eat-2* and *isp-1*.

As suggested by the reviewer, we have revised the manuscript text including the Abstract to indicate clearly the lysosomal aspects that are altered in a certain condition and we have included the point about “lysosome remodeling” in the Discussion section.

6) The authors have employed mutants lacking CUP-5 and CPL-1 activity to conclude that lifespan extension in C. elegans requires lysosome function. However, whether decline in lysosomal function during aging is comparable to what is observed in the cup-5 and cpl-1 mutants is not clear. This can be addressed by functional assays such as cathepsin processing, LSG/LTR ratio tested in any one of the mutants (cup-5/cpl-1) and compared with worms at day 9.

As suggested by the reviewer, we examined the acidity and degradation activity of lysosomes in *cup-5* by LSG/LTR and CPL-1 processing assays. As shown in Figure 9—figure supplement 1P-X, the fluorescence intensity ratio of LSG vs. LTR was reduced significantly in *cup-5* mutants at days 1, 3, 5, while the LSG/LTR ratio in *cup-5* was similar to wild type at day 9. CPL-1 processing was also reduced significantly in *cup-5* compared with wild type and the reduction was observed at all adult ages that were tested (days 1, 3, 5, 9) (Figure 9—figure supplement 1Y, Z).

7) The authors have shown that ratio of lysosensor to lysotracker is reduced in aging worms, indicating that lysosome become less acidic (Figure 2I). Further the pHTomato experiments also indicate that pH is less acidic in aged worms. It would be very informative if the authors can measure the actual pH of lysosomal compartments to understand the magnitude of pH change during aging. If the lysosomal pH measurement is not feasible in this model system, the findings with the pHTomato probe should be complemented with data that directly report pH change such the LSG/LTR ratio. The latter experiment can be added to Figure 6.

The measurement of lysosomal pH is not feasible in live *C. elegans*. We performed the LSG/LTR staining assay as suggested by the reviewer to further support the pHTomato assay. These new data are presented in Figure 6L, Figure 6—figure supplement 1C-J’’, Figure 7L and Figure 8L in the revised manuscript.

8) The discussion on metabolism and role in aging and lysosome function begs the question on what happens to mTORC1 activity. Do the long-living mutants have lower mTORC1 activity? If so, could this help explain lower tubulation in long-living C. elegans. This may be connected to mTOR modulation of lysosome tubulation and remodelling in mammalian macrophages exposed to LPS (Saric et al., 2016; Hipolito et al., 2020). If nothing else, this possibility could be discussed.

The TOR signaling pathway regulates longevity in multiple species and inhibition of TOR extends lifespan in model organisms including yeast, *C. elegans,* fruit flies and mice. The TOR pathway has been shown to link to other longevity pathways such as Insulin/IGF-1 signaling (IIS) and dietary restriction (DR). However, it has not been tested directly whether TOR activity alters in long-lived *C. elegans* mutants. On the other hand, mTOR is important for lysosomal tubulation in the reformation process (Yu et al., 2010), and is required for LPS-induced lysosome remodeling in macrophages and dendritic cells (Saric et al., 2016; Hipolito et al., 2019). As suggested by the reviewer, we examined whether TOR is involved in forming lysosomal tubules in aged *C. elegans* adults. We found that knockdown of the *C. elegans* TORC1 components LET-363/mTOR, DAF-15/Raptor or C10H11.8/mLST8 had no effect on lysosomal tubules in adult hypodermis at day 5, while the length of lysosomal tubules in *let-363* RNAi and *daf-15* RNAi worms increased at day 1 (Author response image 2A-J). Moreover, inactivation of *let-363* by RNAi did not affect lysosome tubules in long-lived mutants *daf-2*, *eat-2* and *isp-1* (Author response image 2K-Q). Thus, TORC1 activity may not be required for age-associated lysosomal tubule formation in worms. We have discussed this point in the revised manuscript.

**Author response image 2. sa2fig2:** Knockdown of *C. elegans* TORC1 components does not disrupt lysosomal tubule formation in aged adults. (**A**) List of TORC1 components that are tested. (**B-I**) Confocal fluorescence images of the hypodermis in wild type expressing NUC-1::CHERRY and treated with control RNAi (**B, F**), *let-363* RNAi (C, G), *daf-15* RNAi (**D, H**) or *C10H11.8* RNAi (**E, I**) at adult day 1 and day 5. (K-P) Confocal fluorescence images of the hypodermis in *daf-2*(*e1370*), *eat-2*(*ad1116*) and *isp-1*(*qm150*) expressing NUC-1::CHERRY and treated with control RNAi (**K, M, O**) or *let-363* RNAi (**L, N, P**). In (**B-I, K-P**), white arrowheads indicate vesicular lysosomes; white and yellow arrows indicate short and long lysosomal tubules, respectively. The length of lysosomal tubules is quantified in (**J** and **Q**). In (**J, Q**), data are shown as mean ± SD. 10 tubules were scored in each animal and 20 animals were scored in each strain at each age. One-way ANOVA with Tukey’s multiple comparisons test was performed to compare all other datasets with wild type treated with control RNAi at day 1 or day 5 (**J**), or datasets that are linked by lines (**Q**). ***P<*0.001. All other points had *P>*0.05. N.S., no significance. Scale bars: 5 µm.

9) Figures 9C and G, the images of NMY-2::GFP do not appear to be representative for the quantification result shown Figure 9I. Further, NMY-2 GFP appears cytosolic in daf2 cup-5 mutant (Figure 9H). It is not clear how the authors have quantified punctae (Figure 9K) in this mutant when so much of the protein appears cytosolic.

We have repeated the NMY-2::GFP experiments and revised the images in Figure 9A-D, G. H. In the revised images, NMY-2::GFP puncta that were scored are indicated by arrows. In Figure 9K and Figure 9—figure supplement 1O, we have included the percentage of worms that do not contain GFP puncta at day 5 (0 puncta). In *daf-2*, 74% of worms have no NMY-2::GFP puncta at day 5, and this number reduces to 36% in *daf-2 cup-5* double mutants (Figure 9K).

10) Total CPL-1 protein levels are increased upon aging in WT (Figures 2N and 4S) but, qPCR results in Figure 5C show reduced cpl-1 expression at day 5. The authors should provide possible explanation for this difference (for instance, comment upon the CPL-1 protein stability).

As pointed out by the reviewer, *cpl-1* gene expression declines but the total CPL-1 protein level increases in aging adults. As CPL-1 processing obviously decreases in aged adults, we reasoned that the increase in the total CPL-1 protein level is probably caused by reduced CPL-1 processing and a decline in CPL-1 protein turnover, both of which are consistent with the decline in lysosome activity in aged worms. We have discussed this point in the revised manuscript.

11) From the data shown in Figures 5, 6 and 7, are there any common lysosomal genes that are upregulated in all the three longevity mutants while a corresponding decrease is observed in older worms. These should be highlighted and discussed in the Results.

As suggested by the reviewer, we have analyzed the lysosomal genes that are upregulated by all three longevity pathways. We found that only 2 out of the 43 lysosomal genes, which are downregulated with age in wild type, are upregulated by all three longevity pathways (Figure 6—figure supplement 1). These two genes, *y105e8b.9* and *lipl-7*, encode beta glucuronidase and lipase-like protein, respectively, but how they contribute to lifespan extension induced by the three longevity pathways requires further studies. On the other hand, the IIS and caloric restriction (CR) pathways seem to target different sets of lysosomal genes (hydrolases in IIS and V-ATPase components in CR), while genes upregulated in the mitochondria-defective mutant isp-1 are mostly shared with the IIS pathway. We have discussed this point in the revised manuscript.

12) The end of the Discussion is a bit abrupt. The authors can probably restate the importance of the findings and also it would be of service to the community to highlight specific caveats and shortcomings of the study to avoid excessive extrapolation in future interpretations.

We found that the lysosome-defective mutants *cup-5* and *cpl-1* are slightly short-lived compared with wild type, and both of these mutations significantly reduce the lifespan in long-lived *daf-2*, *eat-2* and *isp-1* worms (Figure 9L-Q). This suggests that lysosome function is important for lifespan extension. In the last paragraph of the Discussion, we would like to discuss how lysosome function may contribute to lifespan extension and how regulation of lysosome and autophagy, which are both important for longevity, may be coordinated. In the revised the manuscript, we have revised the text to explain the points more clearly and discussed the limitation of our study as suggested by the reviewer.

13) For quantification and statistical analysis of the data, it is not clear to this reviewer if the authors examined all animals for a specific "read-out" in one day or during independent days. For example, the authors state "At least 10 worms were quantified in each strain at each stage" when describing the method to measure tubule length. Were 10 animals all done in one day or over different days? This comment applies elsewhere and it is important to know.

We apologize for not explaining our data more clearly in the original manuscript. For measurement of tubule length, 10 animals were scored in each strain at each day. We have revised the text to clarify this point.

14) For quantitative real-time PCR (qRT-PCR) experiments, it is not clear whether RNA was pooled from multiple worms and if so please state the number.

For qRT-PCR experiments, total RNA was extracted from 20 µl worm pellets and reverse transcribed. We have clarified this point in the revised manuscript.

15) For quantification of NMY-2::GFP intensity experiment, reasoning for using 20 worms for quantification at day 1 and 50 worms at day 5 is not clear.

We have increased the sample size from 20 worms to 50 worms at day 1 in the NMY-2::GFP assay.